# Smart Socks and In-Shoe Systems: State-of-the-Art for Two Popular Technologies for Foot Motion Analysis, Sports, and Medical Applications

**DOI:** 10.3390/s20154316

**Published:** 2020-08-02

**Authors:** Andrei Drăgulinescu, Ana-Maria Drăgulinescu, Gabriela Zincă, Doina Bucur, Valentin Feieș, Dumitru-Marius Neagu

**Affiliations:** 1Electronics Technology and Reliability Department, Faculty of Electronics, Telecommunications and Information Technology, University Politehnica of Bucharest, 061071 Bucharest, Romania; valentin.feies@upb.ro; 2Telecommunications Department, Faculty of Electronics, Telecommunications and Information Technology, University Politehnica of Bucharest, 061071 Bucharest, Romania; ana.dragulinescu@upb.ro; 3Automation and Industrial Informatics Department, Faculty of Automatic Control and Computer Science, University Politehnica of Bucharest, 061071 Bucharest, Romania; gabriela.zinca@stud.fim.upb.ro; 4Mechatronics Department, Faculty of Mechanical Engineering and Mechatronics, Biomedical Engineering and Biotechnology Department, Faculty of Medical Engineering, University Politehnica of Bucharest, 061071 Bucharest, Romania; doina.bucur@upb.ro; 5National Institute of Metrology, 042122 Bucharest, Romania; marius.neagu@inm.ro

**Keywords:** pedar, plantar pressure, sensors, sports applications, gait monitoring

## Abstract

The present paper reviews, for the first time, to the best of our knowledge, the most recent advances in research concerning two popular devices used for foot motion analysis and health monitoring: smart socks and in-shoe systems. The first one is representative of textile-based systems, whereas the second one is one of the most used pressure sensitive insole (PSI) systems that is used as an alternative to smart socks. The proposed methods are reviewed for smart sock use in special medical applications, for gait and foot pressure analysis. The Pedar system is also shown, together with studies of validation and repeatability for Pedar and other in-shoe systems. Then, the applications of Pedar are presented, mainly in medicine and sports. Our purpose was to offer the researchers in this field a useful means to overview and select relevant information. Moreover, our review can be a starting point for new, relevant research towards improving the design and functionality of the systems, as well as extending the research towards other areas of applications using sensors in smart textiles and in-shoe systems.

## 1. General Considerations Concerning Gait Monitoring Systems

Feet locomotion monitoring is essential for medical, sports, and ambient assisted-related applications. For example, in medicine, feet status and feet motion analysis are useful to predict diabetic ulceration of the foot and other disorders. In sports applications, feet locomotion monitoring is used to prevent foot injury of athletes and for coaching. In ambient assisted living, motion analysis helps in training the monitoring systems to detect sudden falls and other risky events [1,2].

Many developed systems measure plantar pressure [3,4,5], which can be defined as the pressure exerted on the human foot skin during various activities performed daily [6]. Most studies performed use this parameter in various applications. There are also a few other studies that used, instead of pressure, the evaluation of temperature [7,8,9,10]. This parameter is closely linked with pressure, but it is sometimes considered more efficient, since it is able to be more accurately measured than pressure [7]. There are commercially available thermistors with both small dimensions and high precision that can be used towards this goal. Temperature can be used, for example, as a precise parameter for assessing the risk of diabetes ulceration, although until now, plantar pressure has been more extensively used for this purpose [7]. In addition, several systems that use both temperature and pressure were proposed [11,12].

Plantar pressure is useful also in other applications [6], such as:footwear evaluation (for determining the efficacy of athletic and therapeutic shoes);athletic training (for optimizing sports achievements);clinical gait analysis (for investigating the walking pattern: normal gait and abnormal walking—toe in, toe out, heel walking, or oversupination);evaluation of foot pathologies (e.g., flat foot, diabetic foot, strephenopodia, strephexopodia).

### 1.1. Sensing Technologies for Gait Monitoring

As concerns the types of sensors for gait monitoring that we have already mentioned above being used by the various in-shoe systems, one can classify these sensors into five main categories [13]:capacitive sensors (composed of two plates, electrically conductive, with an insulating elastic layer between them; when a force is applied, the distance between the plates is modified, and a variation of the voltage is thus produced);resistive sensors (the most used type; they are made of a conductive polymer; when a force is applied, the resistance of the material decreases with the increase of the applied pressure);optoelectronic sensors (composed of a transmitter—generally a laser or a light-emitting diode (LED)—and a receiver—generally a photodiode; between them, there is a silicon-based structure; when a force is applied, it causes a deformation of the cover, the screening of the emitted light, and a proportional variation of the output voltage of the sensor);piezoresistive and piezoelectric sensors (devices that use the piezoelectric effect: the variations of an applied pressure are converted into electrical charge and thus can be measured; for the piezoresistive ones, when the material is stretched, a variation of its electrical resistance takes place and can be measured) and textile sensors (conductive inks are used for creating a textile material that is thin and sensitive to pressure; in this material, one can include a high number of sensors, which however have the drawback of being nonlinear and suffering from a significant hysteresis).

### 1.2. Classification of Gait Monitoring Systems

Currently, there are four types of devices for measuring plantar pressure and for gait analysis [1,2,14,15]:Platform systems (generally embedded in a treadmill):They are considered the gold standard in plantar pressure measurement and possess the advantages of a high precision of measurements and a high spatial resolution. However, they also have the drawback that they can be used only in a laboratory or in a hospital (they lack portability).In-shoe systems:More flexible and mobile, with improved performance and efficiency in terms of circuit solutions, power consumption, and communication technology, with a reduced cost as compared to platform systems, able to measure the distribution of the plantar pressure within a shoe, and able to provide a high number of recorded steps and, thus, a long-term recording of gait, both indoors and outdoors; however, their precision is inferior to that of platforms;Smart wireless insoles:They avoid the drawback of the two previous systems, which have to use electrical wires for sensor connection and for the data acquisition system around the waist. Moreover, as in-shoe systems are not suited for long-term outdoor measurements, smart wireless insoles can be used both indoors and outdoors. They are usually equipped with a data transmission device, such as a Bluetooth or WiFi module, and an energy source, but have the disadvantage of forming an additional elastic layer inside the shoe, which can have a thickness up to several mm and may distort the real data of the foot’s plantar loading. In addition, they are relatively expensive and not suitable for daily use.Smart socks:They are textile-based systems with integrated sensors that avoid the drawbacks above, but have the disadvantage of being handmade or using a complicated fabrication technology.

### 1.3. Features of Foot Motion-Based Systems

Gait monitoring and foot motion-based systems should have several features (Table 1). First of all, they should be wearable and should make little use of cumbersome wiring. To avoid the limitation imposed by wire connections, they should use wireless technologies, thus being able to perform both indoor and outdoor measurements; in-shoe systems are better than platform ones for this purpose. Secondly, they should have pressure sensors with an accuracy and reliability sufficient for being able to pass many repeated loading cycles. Thirdly, they should cause minimal discomfort to the wearer [6]. Lastly, the system should be safe and should be provided with appropriate casing such that injuries are prevented.

As concerns the contribution of the paper, we mention the following:We propose an original definition of smart socks and highlight the most relevant contributions in the field of smart socks and in-shoe systems.We compare the performance of different in-shoe systems and platform systems and consider the main applications in fields such as: medicine, sports, and wellness assessment.We emphasize the challenges faced by these systems and the issues that are still unsolved.

The remainder of the paper is as follows: Section 2 presents the proposed definition of smart socks and overviews smart sock technology. Section 3 details smart socks application in gait analysis, whereas in Section 4, the medical and sports applications involving smart socks were depicted. Section 5 is dedicated to the description of the Pedar system and to the validation and repeatability tests performed for the Pedar system and for other gait monitoring systems compared to Pedar. In Section 6, Pedar medical and sports applications are reviewed and emphasized. Section 7 presents other research initiatives of in-shoe systems. The challenges and the open issues are detailed in Section 8. Finally, Section 9 concludes the paper.

## 2. Smart Sock Definition and Principles of the Technology

This section presents an original definition of the smart sock, together with the advantages and drawbacks of the technology.

The term “smart sock” appeared for the first time in the scientific literature with the significance of a smart device able to acquire motion-related data in 2005 [16].

Smart socks are foot wearable devices that integrate one or multiple sensing technologies, data acquisition and transmission modules, together with the implementation or adoption of data communication protocols to assure the communication between the sock and the devices capable of processing the signals from sensors and visualizing the analyzed data.

A narrower description of smart socks was provided in [17], which states that a smart sock system may consist of a pair of socks, specially designed so as to include pressure sensors, conductive lines, and a block for data acquisition, which is able to communicate with the computer via Bluetooth [17].

Smart socks are not restricted to pressure sensor technology, though. Other types of smart socks comprise EMG sensors to determine the activity of leg muscles around the ankle [18], in order to assess the risk of accidents or the health condition, while others are equipped with more types of sensors for determining the overall score of wellbeing [19] or health disorders [20]. Smart socks are easy to wear, minimizing the discomfort for the subjects that use them [17].

On the other hand, this technology also presents some drawbacks. Firstly, the wearable electronics are only mounted on the textiles, not fully embedded into them. This may present a discomfort to the user, and moreover, normal gait conditions might also be affected. Secondly, this makes the textile structure unstable, thus increasing its susceptibility to noise. In order to obtain the useful information from the signal thus affected by noise, advanced signal processing techniques must be employed. Thirdly, smart socks cannot be washed; for washing, first the electronic components must be removed. This latter disadvantage is under research to be alleviated by designing for the textiles a waterproof enclosure in which the electronic components might be introduced [21,22].

## 3. Description of Proposed Solutions Using Smart Socks for Gait and Foot Pressure Analysis

Various wearable smart sock systems have been proposed in the literature for the acquisition of spatial and temporal parameters of various types of motion (such as walking, running, jogging, etc.). As compared to accelerometer devices (traditionally used in these applications), smart socks possess the advantage of being able to monitor the contact between the feet soles and ground [17], which makes them useful also for measurements of pressure distribution across the foot during walking or running.

This section describes different smart sock designs and solutions targeted to physiological locomotion analysis, plantar pressure measurement, and counting of steps. These designs are reviewed in Table 2.

### 3.1. Smart Socks for Gait Monitoring and Partitioning of Gait Cycle

One of the early models of smart socks for gait analysis was developed in 2013 by an Australian research group, in collaboration with a scientist from Israel [23]. The proposed device was targeted to people suffering from gait disorders. The sock was made from multiple sensor patches. Conductive thread, placed between a neoprene and a conductive fabric, was used for designing each patch. A Secure Digital card was used for connecting the sensors to a data logger. The sensor sock is able to detect important events that may occur during the stance and stride phases of the gait cycle. The F-scan insole (Tekscan, Inc., Boston, MA, USA) was used for validating, by means of comparison, the proposed smart sock system. A 1.6% difference between stride durations and a difference of 3.8% between stance durations, for the smart sock as compared to the F-scan insole, were observed, certifying the potential of the smart sock to be used in various applications [23,24].

A group of researchers from Latvia proposed several systems that use the smart sock in applications where locomotion parameters are analyzed [17]. These smart socks include sensors that, by being knitted in the sock, reduce to a minimum the discomfort to the user. In one of these studies, two types of locomotion were analyzed: walking normally (with a velocity of 3 km/h) and walking at a brisk pace (with a velocity of 6 km/h) while wearing the smart sock system. The five sensors embedded in the sock were placed as shown in Figure 1: Sensors A and B in the front part of the foot, C and D – in the middle, and E – in the rear part (more exactly, on the heel, as stated in [17]). However, please note that in [17], there was a discrepancy between the graphical placement of the sensors (see Figure 1) and the explanations provided.

When the heel strike pattern is present (in walking or running), because the first to hit the ground is the heel, Sensor E is first activated, reaching the maximum value before the sensors placed in the front part of the foot (Sensors A and B). When another type of strike pattern is used when running, Sensor E will reach its maximum value later or at the same time as the other sensors. These considerations were taken into account in order to develop an algorithm for distinguishing heel strike and non-heel strike walking and running modes.

The above design was used for detecting excessive pronation and supination gait conditions that may lead to injuries, both when walking and running. The smart sock has advantages over previously proposed alternatives, being less expensive and more comfortable, and it can be used outside laboratories as well, by non-professionals.

A method was also proposed, in which the values given by the sensors are converted into a pressure vector, which can be further used for describing each step [25]. In particular, this method is capable of detecting, for each step, the excessive pronation and supination of the foot.

The proposed smart sock with five sensors was also used for partitioning the gait cycle. This analysis is useful because of the complexity of the gait, which involves both the central nervous system and parts of the skeleto-muscular system. Therefore, if one or more of these parts are affected by a disease, this is reflected in the gait. Thus, gait partitioning and analysis are important for early diagnosis of diseases, prevention of injuries in sports, and monitoring of the rehabilitation after an operation. Gait partitioning is generally achieved by using insoles and mats that are sensitive to pressure, which are considered the gold standard. An alternative is the use of smart socks, which however have a limitation imposed by the small number of sensors, complicating the task of gait partitioning. The number of phases of the gait cycle generally varies between two (stance and swing) and eight phases.

Another proposed partitioning [26] involved six gait phases, namely: initial contact, loading response, mid and terminal stances, pre-swing and swing phases. As compared to the systems with eight phases, here the swing phase could not be divided into three parts (initial, mid and terminal swing), because in order to distinguish between these three sub-phases, information about the angle between tibia and ground and of the flexion of the knee was not available. The feasibility of the method was verified by an experiment with three participants. The measurement results were compared to the literature data, and the match was very good in most cases. The algorithm for gait partitioning was also described.

A method for gait segmentation was also proposed by researchers from Chile [22]. As opposed to the method described in [26] who used five piezoresistive sensors, this proposal involved a number of eight capacitive sensors in each sock (therefore, a total of sixteen sensors for both feet). The positions of the sensors were chosen as in Figure 2. An algorithm was developed for the segmentation of the gait cycle into phases and subphases and for gait parameters’ determination, enabling the estimation of CoP, velocity, cadence, double support time, and of the time percentage of stance and swing phases [22].

Smart socks have also been recently used for differentiating between normal and abnormal gait. An STS (smart textile system) was proposed that included a smart sock, commercially available (Sensoria), together with a smart shirt. The analyzed conditions were normal gait and four different simulated abnormal gaits. Different methods were used for classification, including SVM (support vector machines), ANN (artificial neural networks), LDA (linear discriminant analysis), and kNN (K-nearest neighbors). The best accuracy (over 98%) of the discrimination between normal and abnormal gait was obtained using kNN, with the other methods also achieving high levels of accuracy [27]. The same research group used three of the previous four methods (the above, without SVM) for discriminating between three different postures (lying down, sitting, and standing) and various walking and running activities, with different speeds, using Sensoria smart socks, a smart undershirt, and finally, a system that combines both of them. The global accuracy obtained for the discrimination was approximately 98% [28].

A smart sock system, named the DAid^®^ Pressure Sock System (DPSS), was developed by the same research group [2], and the accuracy of the proposed system was compared with two other widely used methods (a force platform and an optical system) concerning the measurement of the temporal parameters of locomotion [1]. The developed system (Figure 3) contains an array of piezoresistive pressure sensors, knitted together with electro-conductive lines, along with electronic devices for data acquisition and transmission from sensors to a computer or smartphone, where the data are processed [1]. The placement of the sensors was similar to the one shown in Figure 1.

In order to validate the DAid^®^ Pressure Sock System, the obtained temporal data of locomotion were compared with measurement results from a force platform (BTS P-6000, Italy) and an optical system with LEDs (OptoJump Next, Microgate, Italy). Three types of locomotion were analyzed: race walking, normal walking, and running. For validation, step time was considered, where step time represents the time difference between the moments when the two feet strike the ground, considering the same step. To compare with the OptoJump system, the authors in [1] used Bland–Altman plots. As seen in Table 3, the absolute mean difference between the step times corresponding to each system (smart sock proposed in [1], respectively OptoJump), for all types of locomotion, is sufficiently low to validate DAid^®^. The comparison between the proposed smart sock and the BTS system was made only in terms of the values of the mean ground contact time (Table 4).

The authors concluded that all three devices showed similar step times, thus validating the proposed smart sock system. As advantages of the system, one can mention: it is mobile, lightweight, does not cause discomfort during walking or running, and offers precise temporal locomotion data; therefore, it can be used in medical and sports applications as a practical tool [1].

DPSS was also validated [2] in terms of data repeatability, obtaining a maximum standard deviation of 10% for the signal variation of all eight sensors. This was possible by comparing gait in different walking and running modes, for both normal (asymptomatic) and flat foot. Thus, the tests were done using seven maneuvers: foot up, foot down, normal walking, simulation of supination (under-pronation), simulation of over-pronation, heel running, and toe running. The results showed that the proposed system was able to recognize walking versus running patterns and their long-time alterations and also gave the possibility to control the relative pressure distribution and the temporal gait characteristics [2].

Another validation of DPSS was performed by using a Pedar insole system (described exhaustively in Section 5) as a reference. The smart sock system contained six pressure sensors (similar to the distribution in Figure 1, but with two sensors, instead of one, in the rear part of the sock). The sampling frequency was chosen up to 200 Hz, with the possibility of being reduced, for achieving a lower consumption of energy, for example when longer durations (e.g., more than eight hours) of continuous monitoring are required. The obtained average differences between the smart sock and Pedar were very small: 1.34% (for the stride times) and 1.75% (for the step times), thus assessing the performance of the DPSS smart sock in terms of its temporal characteristics (time-domain signals’ variations). Measurements of the plantar pressure of the foot during running were also done. For the characterization of the acquired measurements, two methods were also developed, specifically for the smart sock system (pressure wave and force vector methods) [32].

Shoe cushioning properties were tested in another application of the smart sock system developed by the same research group. The choice of the shoe with the most appropriate properties is important for people with diseases affecting the feet or with problems of locomotion, but also for other categories such as elderly people and sportsmen. As compared to the previous version of DPSS, the proposed pressure sensors showed a better sensitivity and also a broader working range. Moreover, DPSS is less costly, can also be used outdoors, and avoids the strong underestimation of data as compared to force platforms, which are generally used for the estimation of shoe cushioning. The new sensors were designed and tested, the best sensitivity being achieved by the “curved line”-type sensor [33].

A newer version of DPSS was proposed recently by the same group, together with a novel method for processing, analysis, and representation of gait parameters during outdoor walking and running. The system comprised six sensors and redesigned conductive pathways, such that all connectors between the sock and the electronic device can be placed only on one part (lateral) of the cuff of the socks. Concerning the design of the device, an improvement as compared to the previous DPSS version was obtained by replacing the voltage divider circuit (that converts the resistance of the sensors into voltage) with a solution for measuring the resistance of the sensors by first injecting the stabilized current and then by measuring the voltage drop over the sensor. Thus, the possible range of the resistance of the sensors increased from 0–60 kΩ to 0–1024 kΩ. The sampling frequency was set up to 200 Hz. The proposed method considers that the foot loading during gait is similar to the propagation of a shock or seismic wave, where the reference points are the separate foot sensors. After processing, the data are presented as loading wave plots (WPs), which show feet plantar load propagation during stance/gait. The obtained WPs were compared to the ones obtained by an expert in sports medicine. A good agreement was observed. The method was also demonstrated to simplify the study of gait and the interpretation of the obtained results [14,34].

Other types of smart socks were also proposed and validated in studies. One example of a commercial smart sock is Sensoria (Sensoria Inc., Redmond, WA, USA), which was validated in gait monitoring applications in a study where it was compared to GAITRite (CIR Systems Inc., Franklin, NJ, USA), a clinical system for gait monitoring that was already validated, therefore chosen as the gold standard. Among the twenty-nine participants in the study, fifteen were healthy, whereas the others had a certain neurological disease (Parkinson’s, acoustic neuroma, delay in development, or stroke) that might lead to gait impairment. Three parameters were measured: step count, velocity, and cadence. The results for the first two parameters were without significant differences between Sensoria and GAITRite. Cadence presented differences; however, these were not higher than the standard error of measurement of GAITRite. Therefore, Sensoria smart socks were validated for measuring the number of steps and the velocity, being useful for both healthy patients and the ones with various diseases that might affect normal gait [29].

### 3.2. Smart Socks Applications for Plantar Pressure Measurements

Different research groups proposed using smart socks for plantar pressure monitoring. One such system [35] was targeted towards the detection of abnormal changes of relative plantar pressure values, i.e., persistent pressure changes as the condition of the foot degenerates from normal to abnormal state. By measuring relative and not absolute plantar pressure values (which depend on several factors, such as the measuring method, the dimensions of the foot, and the patients weight), the complexity of the system and of the data analysis was reduced, while still being able to efficiently detect pathological changes in plantar pressure. Furthermore, soft sensors, integrated into the textile material, were developed and used, instead of conventional rigid sensors. During the knitting process of the sock, a pressure sensing matrix with four sensors is integrated into it. Three sensors are placed under the metatarsal heads and one under the heel, these being considered the points that bear the most part of the body weight; therefore, this choice enables the maximization of the abnormal relative pressure detection. This matrix is made from a flexible piezoresistive material. Attached to the lower limb, there is an FPC (flexible printed circuit board), where the control unit of the sock is assembled. When it is not in use, the FPC can be easily detached, which is an advantage over conventional systems with rigid PCBs. The design is battery-free, having as the power source only the smartphone of the user, as compared to other platform and in-shoe commercially available systems, which are battery operated. The smartphone is equipped with an RFID (radio frequency identification) reader unit, and the RF (radio frequency) energy, harvested from it, powers the system. The smart sock includes an embedded sensor-tag, from which the RFID unit is able to read the values of the foot pressure. These values are then digitized and stored in the sensor-tag memory. Thus, the tracking of the values of the relative foot pressures is made possible. Experiments were performed, which confirmed the reliability of pressure measurement and RF energy harvesting.

A small group of normal subjects participated in another smart sock study, where the variations of the CoP (center of pressure) parameter were evaluated both by using Sensoria socks (Sensoria Inc., Redmond, WA, USA) and the gold standard, the Zebris platform. Each sock contained three sensors (placed on the heel, on the first, and on the fifth metatarsal bone) and a three-axis accelerometer. For the posturographic assessment, twenty tests were performed. For validation, from center of pressure (CoP) data, the following parameters were computed: the sway path (SP) calculated as the total length of the CoP path and the mean sway velocity (MV), which is the average velocity that the CoP moved, calculated by means of total distance traveled and dividing it by total time of test. The same SP is automatically calculated by the Zebris system (ZSP), and it was used as the gold standard value. Based on the results of the comparison made between the two systems (Table 5), the authors concluded that Sensoria smart socks are a reliable alternative to Zebris and, moreover, benefit from a lower cost [21,30].

The same research group from Italy developed more recently another prototype of e-textile sock, named SWEET-Sock. The sensor system was composed of three textile sensors (for plantar pressure acquisition) and an accelerometer (for obtaining signals from the lower limb motion). The piezoresistive textile sensors used were EeonTex (Eeonyx Corp., Pinole, CA, USA), made of conductive non-woven microfibers, coated uniformly with doped polypyrrole (PPy). The device also included a voltage divider (for conditioning the detected biosignals), a LilyPad Arduino (for biosignal acquisition), and a Simblee BLE device (for biosignal transmission). The main parameters for postural and gait analysis can thus be obtained [36].

For the recognition of movement patterns, various conductive textiles were compared by an Austrian research group, in terms of their sensing performance when included in smart socks. These socks were obtained from commercial ones, into which small strips of textiles, with piezoresistive properties, were sewn. Two microcontrollers were used, for each sock, for the preprocessing, filtering, and transmission of data. The pressure distribution at protruding points of the foot was used to determine the center of gravity of the body. Then, the distribution and displacement of this center of gravity were used to obtain information on motion. All measurements were stored on an Android phone [24].

Compressible soft robotic sensors (C-SRS) are another type of sensors that were proposed for measuring plantar pressure in a smart sock named GRPS (ground reaction pressure sock). Different movements were performed by thirteen participants in order to evaluate the foot ground reaction pressures using ten C-SRSs (StretchSense, Auckland, New Zealand) placed on top of a BodiTrak vector plate (Vista Medical, Winnipeg, MB, Canada), positioned in turn on a Kistler force plate (Novi, MI, USA). A limitation of the study is that C-SRSs operated at a frequency of only 25 Hz, whereas generally, a frequency of 200 Hz is used when capturing human movement, and a frequency of 1000 Hz is generally used by force plates. An improvement could be obtained by placing the C-SRSs directly onto the socks of the users, thus identifying the center of mass for the metatarsal heads. Despite these limitations, the effectiveness of determining the ground reaction pressures by using C-SRS demonstrates the feasibility of the GRPS system [37].

Other smart socks were proposed that use polymeric optical fibers (POF). A recent study evaluated the ability of inserting POF in a textile material and to measure, during walking, not only the pressure, but also friction [38]. The mechanical properties of the fibers, such as bending rigidity, elongation, and minimum bending radius (before a plastic deformation occurs), were used for comparing various POF, in order to choose the most appropriate to be inserted into the fabric. Afterwards, the POF thus chosen was introduced into three different types of fabric, and its sensitivity to friction and compression was demonstrated. The structure possessing the highest sensitivity was selected for fabricating the smart sock prototype. Thus, POFs can be used as pressure and friction sensors in the smart sock. A feasibility study was finally performed, using three optical fibers inserted in a sock, in three different zones, demonstrating the possibility of detecting the different phases of walking, while allowing maintaining the walking gait and measuring the stresses exerted on the fibers in different zones. As shown in Figure 4, Guignier et al. presented the knitted sock to evaluate the capability of the optical fibers to give insights into the friction and pressure during walking.

However, there are still some drawbacks concerning the use of optical fibers in smart socks. The most important one is the fragility of the optical fiber. Thus, despite the accuracy of the sensors verified during laboratory tests, the applicability of optical fiber-based smart socks is limited [9].

### 3.3. Smart Socks for Counting Steps

Counting steps is another application where smart socks have proven to be useful. Currently, the steps are generally counted by using pedometers, which may be obtrusive and also have the disadvantage of being unable to differentiate between activities, or by using pressure sensor pads, which are sometimes too costly and cannot be used everywhere and anytime, because of their limited size. Smart socks overcome these shortcomings, being able to classify physical activities with a high accuracy and also being a lower price unobtrusive device without limitations of time and place for its use.

One proposed system for counting steps consisted of smart socks that gather information concerning motion and the degrees of ankle bending and send it to a computer and a control software, which includes three algorithms: for classification, step counting, and interaction with the user. In turn, the smart socks contain seven sensors: four attached in the ankle position (for bending assessment) and three embedded in a box attached to the front of the sock (an accelerometer, a magnetometer, and a gyroscope). For data recognition, a classification algorithm based on SVM was developed, which is able to discern the type of the activity that the user is performing at a certain time. For the activities classified as walking, the system is able to provide the total number of steps of the user, with a high accuracy [39].

In another study, steps were counted for slow walking with different speeds, in healthy adults. The steps were counted using three different methods: using a smart sock worn on the left foot, a pedometer (from Omron), and with a pedometer included as an application (from Cross Forward Consulting, Herndon, VA, USA) in a smartphone. The smart sock proved a higher accuracy as compared to the two employed types of pedometers, when counting steps in walking at slow speeds and when walking on a treadmill (Quinton Q-Stress, USA) [31].

Another research group investigated methods for finding frailty phenotypes that can be measured not only in clinics, but also in unsupervised settings. One-hundred sixty-one older adults participated in the study. Their gait performance was analyzed, and data were gathered from sensors in order to determine the gait parameters and to discriminate various stages of frailty. An algorithm was proposed for predicting these stages by using the gait parameters. Its accuracy was evaluated by an artificial neural network model. Wherever the sensors were located, the model proved to maintain the same high accuracy in predicting the frailty stages. It was demonstrated that a single gyroscope is sufficient in order to extract all parameters needed. The gyroscope overcomes the disadvantages of accelerometers, which are sensitive to the sensor location, and also of pressure sensors, which need to be calibrated regularly. The proposed algorithm could be implemented, as the authors stated, in Sensoria smart socks, in smart shoes, and others, in which the gyroscope sensors can be integrated [40].

## 4. Description of Proposed Methods Using Smart Socks in Health and Wellness Monitoring

This section emphasizes the smart socks developed as a solution to medical and wellness applications.

Table 6 reviews the main research initiatives concerning smart socks systems and their applications, whereas in Table 7, the main commercial smart socks and their applications are emphasized. Each system is presented from the point of view of application, the general method used to respond to the corresponding application, and the sensing technology. Moreover, where possible, the communication technology is emphasized. Some authors refer, for example, to wireless communication, without mentioning the communication technology (CoT) they used. Consequently, in the table, for such systems, the note “no CoT mentioned“ will be attached.

### 4.1. Smart Socks for Patient with Diabetes

As seen in Table 6 and Table 7, smart socks have found various applications in medicine. One such application is for patients with diabetes. Repetitive elevated stress, misfitting footwear, objects inside the shoe, and others may contribute to excessive pressures and create foot ulcers in these patients. Therefore, a device is needed to monitor the external pressures involved. In 2011, in France, two research centers (TIMC-IMAG, Techniques de l’Ingénierie Médicale et de la Complexité — Informatique, Mathématiques et Applicationsand AGIM, AGeing and IMagery), in collaboration with a clinic (Centre de l’Arche) and two companies (TexiSense and IDS) announced the future design of a smart sock capable of monitoring these pressures. This was called the TexiSense Smart Sock [41]. Three years later, in 2014, the design and test of this smart sock were reported. This fully wireless device was composed of three parts, as emphasized in Table 8.

Eight pressure sensors were knitted in the sock. During gait, the foot pressures were measured and transmitted, by means of a Bluetooth connection, to a smartphone [42,43].

Smart socks can be used for diabetic patients, not only for monitoring the pressure, but also the temperature. Various designs have been proposed for temperature monitoring with smart socks, both for diabetic and for healthy patients. Concerning the diabetic ones, temperature sensor integration technology in smart textiles [55] and TSS (temperature-sensing socks) technology was proposed, which is able to detect temperature variations in the feet. These temperature variations are good predictors of foot ulceration in diabetics, a condition that can lead to complications, even to amputation. Such a TSS including a temperature-sensing yarn, with nano-sensors glued in the yarn made of polyester copper, a box for data processing (wireless communication), and a battery energy supply was proposed in [8]. The sensors acquire the body temperature data, then these data are transmitted by the communication system to a smartphone application, where they are stored and analyzed.

A prototype smart sock was also proposed, using temperature sensors for monitoring the foot temperature levels of the patients. Tests were performed using LM35 and NTC (negative temperature coefficient) thermistors. The device aims to offer the patients a continuous assessment of their health conditions, enabling them to have information in real time concerning foot temperature levels [7]. Another research group reported in the same year, for the first time, a smart sock wireless device that achieved a continuous monitoring of the feet temperature. The sock can be worn by patients with diabetes and neuropathy on a daily basis. The chosen textile materials with sensors were the Siren Diabetic Socks (Neurofabric, Siren Care Inc., San Francisco, CA, USA). There are six sensors, placed in various positions at the bottom of the foot, that take measurements of the temperature at every 10 s and can detect any abnormal increase of this parameter. The information can be sent by Bluetooth to the mobile phone of the users. The proposed system is able to detect any warning sign of a possible ulcer of the feet in diabetic patients [9].

Such systems were also proposed for healthy people (also exposed to risks of developing diabetes). In one report, the skin temperature of the feet, together with the plantar pressure, were measured by using smart socks. Moreover, the study aimed to assess the relationship between an increased skin temperature and a plantar pressure overload as predictors of foot ulceration. However, the participants in the study were not diabetic patients, as in the previous studies, but healthy subjects. Seven thermal sensors (10 K NTC thermistors) were woven into the socks’ fabric. The upper part of the sock was connected to a central unit (composed of an Arduino Mega development board, a screen, and a battery), having the role of converting variations of the resistance of the sensors into temperature variations, which can be seen on the screen. The results of the study showed a correlation between foot temperature variations and the distribution of the plantar pressure. Furthermore, for predicting changes of this latter parameter, it was found that two of the six sensors were sufficient [10].

Another proposed smart sock used in medical applications is SmartSox, with highly flexible and thin (<0.3 mm) optical fiber sensors, based on FBG (fiber Bragg gratings) embedded in a standard sock. The prototype was designed and fabricated by Novinoor LLC (Wilmette, IL, USA) and allows the simultaneous measurement of temperature, plantar pressure, and the range of toe motion. The obtained data are transmitted to a microprocessor for storage and monitoring. The validity of this wearable technology was demonstrated for the first time in [50]. The device was applied for managing the biomechanical risk factors in patients suffering from DPN (diabetic peripheral neuropathy).

### 4.2. Smart Socks for Periodic Limb Movement Disorder

Smart wearable socks were also proposed and prototypes designed for a home system that monitors periodic limb movement disorder (PLMD)—a sleep disorder in which periodic limb movements (PLM) induce nocturnal awakening of the patient. Until recently, the only practical method for PLMD screening was PSG (polysomnography), which is time-consuming and costly, requiring hospitalization of the patient. In order to solve these problems, a PLM monitoring system was proposed [44] that does not require hospitalization, but can be used at home. The solution consists of a prototype sEMG (surface electromyogram) measurement system, including remodeled ready-made stretchy socks and Nishijinfabric electrodes, fixed by the pressure of the fabric adapter and thus preventing the slipping of the electrodes, and therefore, measurement faults due to this cause. The prototype is also able to make a clear distinction between voluntary movements and PLMs, which is an important advantage over other screening devices (that do not have medical approval), which use only an accelerometer for detecting PLM and have a high rate of false detections. The proposed system also has the advantage over conventional electrodes that it can be used by patients without any medical knowledge, who simply need to wear the device as they would normally wear socks.

The same research group also tested such a device on one patient with PLMD, at home, and concluded that the performance of this smart sock system is better than that of conventional devices based on acceleration [56].

### 4.3. Smart Socks for Fall Risk Detection

A prototype of a smart sock based on sEMG was also developed, by another group, for the assessment of the contraction of gastrocnemius-tibialis leg muscles, that offers information on fall risk (described in the report as a case study), anomalies of gait and posture due to aging, sarcopenia pathology detection, etc. Thus, by a continuous monitoring of the electric potentials that the muscles produce, abnormal events might be detected rapidly. Biocompatible electrodes were used, based on HPe (hybrid polymer electrolyte), instead of previous solutions, based on pre-gelled electrodes that could be used only once and produced skin redness. All the electronic components and the electrodes were integrated into the device. The obtained prototype is less invasive, lightweight, less expensive, and easier to appropriately position, as compared to previous models that do not employ smart socks [46].

### 4.4. MONARCA System: Smart Wearables and Sock for Bipolar Disorder

Another disease for which smart socks can be used is bipolar disorder. For such patients, a wearable system, named MONARCA, was designed [20], having as the aims the early recognition of warning signs and the prediction of depressive and manic episodes. The proposed sensor network is centered on the smartphone (which performs the most important part concerning sensing, containing three sensors: GPS (global positioning system), accelerometer, and magnetometer, and enables direct connection between the doctor and patient), also including a wrist-worn sensor (WWS) that detects the movement of the user (and contains another accelerometer and also a gyroscope) and a smart sock. The latter provides pulse signals (that indicate the stress level) and GSR (galvanic skin response). Since GSR signals can be reliably collected only at the feet or palms (the latter being unpractical), the choice of the smart sock is very appropriate. The proposed device is minimally invasive, by combining two approaches: embedding all sensors in a garment and placing as much sensors as possible in equipment that can be worn during daily activities.

### 4.5. Smart Socks and Sensory Augmentation for Prosthetic Limbs

Smart socks have also found interesting medical applications in the sensory augmentation of prosthetic limbs. Such a solution is proCover [47], which is based on smart textiles, enabling amputated persons to “feel” again with that limb. The prototype is non-invasive and can be applied by the patients themselves over the lower limb. The sock consisted of three different layers of fabric, worn one over the other by the patient on the prosthesis. The sock prototype included 192 sensor interactions (16 rows and 12 columns). The sensors change their resistivity when a mechanical force is applied on them. This research was the first to consider the application of textile-based sensor socks for prosthetic applications. The device was successfully applied to patients that were able to differentiate between touches in various places of the leg and with various applied pressures.

### 4.6. Smart Socks for Parkinson’s Disease Patients

Parkinson’s disease patients are another category of people that can be aided by wearing smart socks. A system, which can also be used in other areas of healthcare and also in sports, was proposed in collaboration with several research groups [48]. They developed a cotton sock, both self-functional and self-powered, that uses a hybrid mechanism: piezoelectric—based on PZT (lead zirconate titanate) material—and triboelectric—based on a PEDOT:PSS-coated TENG (triboelectric nanogenerator). This smart sock is capable of energy harvesting and also of sensing different parameters, such as gait, sweat level, contact force, etc.

For patients with Parkinson’s disease, various technologies have been proposed for gait analysis, together with different algorithms, for the diagnosis and monitoring of symptoms. Only a few of these technologies and algorithms, however, have been proven useful in clinical settings, and none has already been validated and used on a large scale [57].

### 4.7. Smart Socks for Patients Who Have Suffered Stroke Events

A proposal for gait abnormalities’ detection in patients with strokes, for their remote assistance, is the so-called MagicSox system, developed by a group of researches at the University of Rhode Island (USA) [49]. The architecture of the system contains a FlexiForce A201 (Tekscan) piezoresistive pressure sensor (on the heel), two flex sensors (one on the heel, the other one on the ankle, on its anterior side), a gyroscope and an accelerometer, both integrated in an Intel Curie processor module included in an Arduino 101 board, and a smartphone, used for collecting data through Bluetooth. MagicSox enables the differentiation between healthy feet movements and drop foot ones. As compared to previous solutions, the proposed system is able to offer more detailed information concerning the gait cycle, foot posture, and ankle flexion.

### 4.8. Smart Socks for Baby Monitoring

Baby monitoring is another application where smart socks can be used. Two such smart sock technology-based devices that are already functional are OSS (Owlet Smart Sock) and Baby Vida [52]. Baby monitoring systems have attracted much controversy. Proponents consider advantages such as obtaining important data about the baby’s health, alerting in time about possible crises, whereas critics stressed the high number of false alarms, the high price of the monitors, the rise in parents’ anxiety, and overdiagnosis risks. Moreover, there has been a long debate (since the early 1970s, when pulse oximetry and cardiorespiratory monitors in home settings were introduced) concerning the benefits and drawbacks of these devices. In 2003, the recommendations made by the AAP (American Academy of Pediatrics) Committee on the Fetus and Newborn discouraged the use of these devices for prevention of SIDS (Sudden Infant Death Syndrome) or for healthy babies, born full-term, expressing concerns because there are no proven benefits and restricting their use to the newborns exposed to high risks [51].

Concerning the aforementioned OSS technology, the largest experiment of home cardiorespiratory monitoring was done by [51], in which 47,495 newborns were monitored for six months, 4.5 days per week, 9.9 h per day, evaluating parameters such as SpO${}_{2}$ (oxygen saturation), HR (heart rate), and AOP (apnea of prematurity). SpO${}_{2}$ and HR (the vital signs) were monitored by using a sensor-embedded sock at the homes of the newborns, during their sleeping. The OSS kit contains three sizes of socks (for ages up to 18 months), a base station, a pulse oximeter, and chords for charging. The sensor sends the data to the WiFi-enabled base station by means of BLE (Bluetooth Low Energy). The base station is also the primary notification system; when HR levels exceed or are under the limits or when SpO${}_{2}$ levels change, it emits visual and audible alert signals [51].

Another work, in the same area as above, was recently performed by a Romanian research group. They designed and implemented an innovative pulse oximeter for newborn monitoring. The system, named P-SOCK, can be used both in NICU (neonatal intensive care units) and at home, by parents, coming at a reasonable price. The idea was to redesign the classic pulse oximeter, under the shape of a sock, being made of a special elastic material. The proposed system consists of two modules. The first one includes an Arduino Nano, a Bluetooth module, and the sock and, inside, a MAX3100 sensor for measuring the heart rate and the oxygen saturation in the blood. The second module consists of a large LED screen, for displaying the values measured by the sensor, and an alarm, for the cases when the normal limits of these values are exceeded [58].

### 4.9. Upper-Limb Smart Textile for Sports Applications

There are also other specific domains of application for smart socks, besides medicine. In sports, these devices are still waiting to be implemented for helping players. One sport in which a recent application was already developed is basketball. This time, it is not precisely a smart sock, but a similar device, to be worn on the hands of the players: a smart basketball glove (SBG). The knitted sensors used were as the ones previously used by the same research group in applications with smart socks and smart shirts. The high speed of response and tactile sensitivity of the sensors suggest that SBG might analyze and monitor in real time the movement of the fingers and wrist and also estimate the forces created by the interaction with the ball of the fingers. This smart glove is useful for analysis and training concerning basketball shots [59].

## 5. Description of the Pedar System, Together with Validation and Repeatability Tests for Pedar and Other In-Shoe and Platform Systems

The Pedar^®^ mobile system is currently considered the gold standard among in-shoe systems and is one of the most used and well-established sensor insole pressure measurement systems [60,61,62,63,64].

The Pedar^®^ mobile system is a matrix insole system, including a dielectric material and, on either side of it, fixed metal strips (horizontally, in rows, and vertically, in columns); a capacitive sensor is placed at every point of intersection between these rows and columns. In order to ensure insensitivity to the humidity inside the shoe, the sensors are placed between two polyethylene layers, and artificial leather covers them on both sides. The number of capacitive sensors on each insole is 99, placed equally on the entire insole area. The thickness of the insole is approximately 2 mm [65].

An important characteristic of an in-shoe system is its repeatability, which can be defined as the ability of an instrument to measure the same values, for repeated measurements in the same conditions, on the same subjects, and using the same experimental devices [66].

In a study for assessing the repeatability of the Pedar system, the foot was divided into 10 regions, as shown in Figure 5: 1—heel; 2—lateral mid-foot; 3–7—first to fifth metatarsal; 8—hallux; 9—second toe; 10—Toes 3–5. To prevent confusion in Figure 6, as compared to [67], we used the appropriate term “lateral mid-foot”, instead of mid-foot.

In order to evaluate the repeatability, six clinically relevant parameters (Table 9) were chosen out of the 18 parameters for which the Pedar software is capable of calculating information.

The conclusion was that the Pedar system was repeatable and the determined pressure values in normal subjects may provide a reference range in clinical practice [67].

The Pedar system was compared to other different in-shoe systems. One such study compared Pedar with a recently designed smart insole (SI). The investigation was done by means of a force plate. The innovative footwear-based smart insole system was designed by the team of researchers in 2015 and validated in comparison with Pedar [61] in terms of CoP accuracy. The results of the validation stage are presented in Table 10.

Pedar was also compared with different in-shoe systems that are already commercially available and validated, such as Medilogic, Tekscan, Zebris, AMTI, and OpenGo. In one such study [69], the Pedar system was compared to two other commercially available in-shoe systems for pressure measurement (Medilogic and Tekscan). The purpose was to evaluate the validity of the three in-shoe systems across a range of magnitudes and durations of the applied pressure, because a similar analysis existed only for plantar pressure measurement plates, but not also for in-shoe pressure devices. All three systems proved a high repeatability for all analyzed variables. As for the validity, the highest one for pressure was exhibited by Pedar, whereas for contact area by Medilogic. Concerning the clinical risks due to in-shoe pressure, the results of the study recommended the Pedar system. Because various devices have different features (e.g., number of sensors, type of sensor technology used, measurement range, etc.), the study also compared the most important features of the three in-shoe systems. Table 11 presents this comparison.

Pedar was also used for a comparison with Zebris FDM-THQ (Zebris Medical GmbH, Isny im Allgäu, Germany), an instrumented treadmill for which the reliability and validity needed an assessment. The reliability of Zebris proved to vary from acceptable to excellent for two of the examined parameters (contact and flight times), but Zebris underestimated a third parameter (maximum vertical force) as compared to Pedar at higher speeds of running [62].

Another research group performed a comparison of measurements of the in-shoe CoP, done with the Pedar-X insole system and with AMTI (Advanced Medical Technology Inc., Kirkland, WA, USA) force plates. The Pedar-X system contained two in-shoe insoles, 2.5 mm wide, each with 99 pressure sensors of the capacitive type and connected to a unit, attached with a belt to the waist of the subject. The data are transmitted wirelessly by the unit to the computer having Pedar-X software installed. This software is able to calculate the in-shoe CoP [74].

The Pedar-X in-shoe system was also compared in another study [60] with the AMTI force plate system (as above) and also with the newly developed OpenGo in-shoe system (Moticon GmbH, München, Germany), for the experimental validation of the latter. The authors of the study reviewed previous attempts of assessing the repeatability of and validating sensor insole systems and considered that they all had the disadvantage of analyzing exclusively walking and running tasks. Therefore, in order to improve such tests, the authors added jumping, body balance, and special motions that imitate cross-country skiing, with simultaneous measurements of all three systems.

The validity and reliability of OpenGo were evaluated in another study [75], for the measurement of temporal gait parameters (gait cycle time, stance time, and cadence), when walking, being compared to an AMTI OR6-7 force plate and a ForceLink instrumented treadmill. The OpenGo system contained, on each insole, 13 capacitive pressure sensors (covering 52% of the insole area), together with an accelerometer, a temperature sensor, and a chip for data storage. The participants wore OpenGo between two pairs of socks (their own socks and thin cotton ones, offered by the researchers), without shoes. Data were collected at 50 Hz. From the results, OpenGo was proven valid and reliable for measuring the aforementioned parameters.

More recently, the OpenGo system was used for unsupervised gait retraining [76], targeting knee osteoarthritis patients. However, the participants chosen for the study were healthy subjects, who performed firstly gait training in the lab, which was then continued in an unsupervised manner, outside the lab. OpenGo insoles were used in the experiment. Data concerning plantar pressure were collected at 100 Hz. According to the obtained results, the system proved to be suitable for gait retraining applications.

Another system (this time a platform one instead of in-shoe) for which repeatability tests were performed recently is the EMED^®^-SF platform system [66], consisting of a floor-mounted, rigid, and flat array of pressure sensors, arranged in a matrix configuration, with clinical applications for gait analysis, diabetes mellitus, hallux valgus, etc. [6,15,77]. This plantar pressure measurement device is able to measure parameters such as: peak pressure, mean pressure, and pressure-time integral. However, the authors observed the lack of indications concerning the repeatability of EMED^®^-SF measurements for the following four types of clinical plantar pressure measurements: static (test performed while standing), static with load, dynamic (when walking normally), and dynamic with load. Therefore, the study performed all four types of measurements in order to assess the repeatability of the EMED^®^-SF system. The procedure was applied on one subject (the carried load being of 1.5 kg), and experimentally, it was found that it can also be applied on others with the same results, demonstrating in this way the repeatability of the EMED^®^-SF system.

The EMED^®^-SF system was used for helping neuropathic diabetic patients assess the risk factors of the recurrence of foot ulcer [78]. The EMED^®^-X pressure platform (Novel, München, Germany) measured the values of the plantar pressure, barefoot and with footwear, with the aid of sensors, four of them placed on each cm${}^{2}$, at a sampling rate of 50 Hz. It was demonstrated, for the first time, the role played, in the recurrence of foot ulcer, by the in-shoe plantar pressures and that the risk of this recurrence can be reduced by more than 50% by an effective offloading as recommended in the study.

Another application of EMED^®^, this time together with Pedar, is for evaluating the possibility of diagnosing anaerobic (non-oxidative) power and capacity [79]. The Bosco test of repetitive jumps was performed in order to determine the anaerobic capacity. The EMED^®^ system was used with Pedar placed in the shoe of the tested participants. The dynamic parameters were recorded at the frequency of 100 Hz. The results confirmed that anaerobic power and capacity can be diagnosed with the proposed method.

Repeatability was also assessed for another platform system, Footscan [80]. For this purpose, five parameters were recorded: contact area (CA), contact time (CT), peak pressure (PP), pressure-time integral (PTI), and maximum force (MaF), in ten foot areas: medial and lateral heel, midfoot, Metatarsals 1–5, hallux, Toes 2–5. The Footscan system (RSscan International, Olen, Belgium) contained 16,384 resistive sensors, with two sensors per cm${}^{2}$, with a frequency of data acquisition set to 125 Hz. The Footscan system proved to be repeatable, being a reliable device for measuring the distributions of the plantar pressure in dynamic conditions.

## 6. Description of Proposed Methods Using the In-Shoe Pedar System

The Pedar system has found applications that might be classified as follows: in medicine, sports applications, and various others. Most of the applications are in the medical field and in sports, whereas other usages are specific, concerning the measurement of certain parameters concerning walking, running, or gait in general. Below, we shall analyze the applications reunited under these three aforementioned categories.

### 6.1. Pedar in Medical Applications

One of the medical applications of the Pedar system is for patients with diabetes and a plantar forefoot ulcer [81]. In order to reduce plantar pressure in these patients, an effective intervention is the total contact cast (TCC) (a well-molded, non-removable cast that is in contact with the entire plantar aspect of the foot and lower leg and protects wounds from further injuries, in order to prevent amputation in patients with plantar neuropathic ulcers), which is able to bear approximately 30% of the plantar load. The highest load is carried by the anterodistal and posterolateral-distal regions of the lower leg. Therefore, for these two regions, the study [81] proposed a method to measure them directly. For this purpose, two capacitive sensor strips (90 cm${}^{2}$) were used for evaluating the TCC wall load (peak pressure, max force, and contact area), and a capacitive sensor insole (Pedar), placed inside the TCC, was used for measuring the plantar load (the same three parameters as for the TCC wall load). The Pedar capacitive sensor insole, having a resolution of 1.2 sensors per cm${}^{2}$, was placed between the plantar surface of the foot and the material (made of cellular urethane) inside the TCC. The magnitude of TCC wall load was calculated.

Diabetic patients were also tested in another study that measured, using Pedar-X, the distribution of the plantar pressure, both when standing and walking [82]. The participants wore EVA (ethylene vinyl acetate) insoles with two different densities, and the effect on the pressure distribution was assessed. The effects were found to be stronger when walking, than in static conditions. Another research involving patients with neuropathic diabetic foot ulcers was conducted in order to evaluate the differences between healing and non-healing ulcers from the point of view of cumulative plantar tissue stress [83]. In order to calculate this stress at the ulcer location, two parameters were measured: daily stride count and dynamic plantar pressure, a progress from previous studies where only peak plantar pressure was considered, without taking into account the role of the ambulatory activity of the patient in healing. In order to measure the plantar pressure, the Pedar-X system was used, with four insoles of different sizes, adapted for each size of shoe of the participants. The sample frequency employed was 50 Hz. No statistically significant differences were found, between healed and non-healed ulcers, in terms of the analyzed parameters.

Osteoarthritis (OA) is another disease where Pedar-X systems are commonly used. This medical condition affects a large number of people. For patients with OA of the ankle (AOA), a study assessed gait symmetry by means of measuring a number of 46 gait parameters and motions relative to the sub-region of the feet. Significant asymmetries of gait were found in one third of these measured parameters and forefoot relative motions. The results were obtained by using an AGA (ambulatory gait assessment) system, composed of Pedar-X pressure insoles and five inertial sensors, with gyroscopes and accelerometers (Physilog, Switzerland), that allows patients to walk in a natural way during the process of gait assessment [84].

In another research work [85], this time for patients with OA of the medial knee, the relationship between three factors was analyzed: knee pain, plantar forces present in the shoe, and the static posture of the feet. Pedar-X was used for measuring in-shoe plantar foot forces when walking, in the shoes the patients commonly wore. Seven parameters were determined using Pedar: three forces (for the lateral, medial, and whole foot), the ratio of the forces corresponding to the medial versus the lateral foot, and three other parameters (velocity of walking, duration of stance, and arch index). The authors concluded that, by redirecting pressures from the medial to the lateral plantar surface (by using footwear, orthoses, etc.), knee pain might be reduced in OA patients.

Furthermore, for patients with knee OA, a weight-supported device was proposed for assistance when walking. Because of the knee pain, they walked with difficulty. After cartilage is implanted into these patients, the weight on the knee joint must be reduced. Thus, a device was proposed that reduces this weight, unloading two thirds or more of the body weight on the knee, enabling the user to walk without a significant change in the gait when wearing the device. For calculating the load values and also four gait parameters (walking speed, step time, symmetry index, and stance percentage), derived from the pressure values, the Pedar-X system was employed, with the sampling frequency set to 100 Hz. Both a basic test and a prototype model were developed, in order to confirm the feasibility of the system [86].

Knee OA patients might also benefit from knee unloading shoes, proposed for reducing the external knee adduction moment (KAM). However, since the effects of this footwear on regional plantar forces were still unknown, a research group tested participants wearing knee unloading shoes and conventional shoes, respectively, and found a lateral shift in plantar pressure and force patterns for the unloading footwear, as compared to the conventional ones. Regional plantar pressures, for both feet, were measured by using the Pedar-X system, with a sampling frequency of 100 Hz. The authors also observed, in 20% of the participants, adverse effects of unloading shoes, which caused foot pain. Thus, this footwear might amplify foot problems in some patients and should be used with caution [87].

Another study [88], also involving patients with knee OA, measured the plantar pressure during walking, as a basis for deciding which footwear and foot orthoses to choose. Three parameters were measured: PP (peak pressure), MaF (maximum force), and CA (contact area), in seven regions underneath the foot: heel, midfoot, first metatarsophalangeal joint (MPJ), second MPJ, third to fifth MPJ, hallux, and lesser toes, respectively. The study had the purpose to be the first to evaluate plantar load differences between elderly women with and without medial knee OA during walking. The Pedar-X system was employed for measuring the plantar pressures, at a sampling rate of 50 Hz. The study found a higher plantar loading (in terms of MF and PP) for elderly women with medial knee OA as compared to those without knee OA, at the midfoot and at the first and second MPJ.

The Pedar-X system can also be used for patients with low back pain (LBP). In a recent study, the effects on neuromuscular trunk responses to sudden perturbations of gait were assessed in female patients as compared to males. Since LBP appears more frequently in females, the neuromuscular differences as compared to males were evaluated. The results were obtained using a Pedar-X plantar pressure insole in experiments where the patients walked on a treadmill, and after the initial heel contact, right-sided and left-sided perturbations were applied. Only the right-sided ones were analyzed, because the plantar pressure insole was used only in the right shoe, directly triggering the perturbations. The obtained results may have a relevance in developing intervention strategies, specific for female and male patients, respectively [89].

The plantar pressure measuring system Pedar-X-32 (Novel GmbH, München, Germany), in the form of a 260-280 mm long insole, was used in a study of gait, in which paraplegic patients (PPs), who are helped to walk by using various gait-assistant aids, participated in tests with a weighted vest (WV) (Healthway, Korea) (fitting a block of steel in the front and back side), which was chosen for the load stimulus. The purpose of the study was to identify static and dynamic balance with the addition of WV for the rehabilitation of PPs. Weight ratios were set as 0%, 10%, and 15% of the weight of the ten PPs who participated as test subjects. Pedar was used in the static balance tests for measuring the CoP, its excursion and velocity. The TUG (Timed Up and Go) test was used for identifying the dynamic balance. Results showed that WVs are more useful than non-weighted ones for rehabilitating PPs; however, no difference was observed between 10% and 15% ratios. WV exercises are expected to be a foundation for methods that maintain and improve the balance ability of PPs, with the final goal of their rehabilitation [90].

Pedar-X was also used for medical shoes’ comparison. A hindfoot relief shoe (HRS) and a hindfoot relief orthosis (HRO) were analyzed in a study and compared to a neutral shoe in terms of plantar pressure changes. The values of the peak pressure were collected from four plantar regions: hindfoot, midfoot, forefoot, and Metatarsal I-V. Data were collected using the Pedar-X system, with a sampling rate of 50 Hz. Obtained plantar pressure values were significantly lower in the hindfoot and metatarsal regions when using HRS and HRO as compared to normal shoes; significantly lower for HRS and slightly lower for HRO in the forefoot region, but significantly increased in the midfoot, suggesting caution especially for patients with midfoot injuries or other impairments [91].

### 6.2. Pedar in Sports Applications

Just like in the medical domain, also in sports, the choice of the right type of shoe is essential. Several studies compared different sports shoes in terms of their performance, by means of plantar pressure determination using the Pedar-X system. Thus, one such study [92] determined the effects on in-shoe pressure determined by rigid and flexible rocker profiles, generally employed for diabetic foot ulcer prevention. For ensuring a proper offloading and a restricted plantar flexion of the toes, usually rockers are stiffened. In order to assess the difference in restricting also toe dorsiflexion, between flexible (allowing dorsiflexion) and rigid rockers, plantar pressure data were gathered from a shoe in five different cases: flexible rockers with the apex positioned at 50%, rigid rockers with the same apex positioning, flexible and rigid rockers, respectively, with the apex positioned at 60%, and a control shoe, without any rockers. Female adults participated in an experiment where, for seven foot regions, three parameters were determined: PP (peak plantar pressure), MMP (maximum mean pressure), and FTI (force-time integral). For measurements, Pedar-X insoles were used, for both feet, with a sampling frequency of 100 Hz. Rigid rockers proved to reduce significantly the plantar pressure in the forefoot region, but also to increase this pressure at the first toe, as compared to flexible rockers [92]. The effect of footwear, together with the one of the stride length, was also assessed in terms of the probability of failure in running (stress fractures) and metatarsal strains. During the stance phase of running, high strains are experienced by the metatarsal bones, making them prone to fractures caused by stress. In order to reduce this risk, special footwear and the reduction of stride length were two proposed approaches.

In a study, plantar pressure values were collected from participants, wearing two types of shoe: traditional and minimalist, at two different stride lengths: preferred stride length (PSL) and 90% of PSL. Pedar-X insoles (Novel, Minneapolis, MN, USA) were used for pressure sensing, in the right shoes of the participants, capturing data concurrently at 200 Hz, whereas in the left shoes, non-functional Pedar-X insoles were placed. The Pedar-X insole was segmented, using the method previously proposed in [67], into four regions: I-V metatarsal, hallux, second toe, III-V toes. The study found that recreational runners, undergoing an acute transition to minimalist shoes, were exposed to a significantly higher probability of stress fracture and increased metatarsal strains. The technique of reducing the stride length with 10% did not prove to be either beneficial or detrimental in lowering the risks of metatarsal fractures [93].

The Pedar Mobile System was also used recently for examining the way in which the hardness of the shoe insole influences the distribution of the plantar pressure in sports applications, particularly in basketball-related movements. Four such movements were analyzed: running; maximal forward sprinting; maximal 45° cutting; lay-up. The reason behind this choice is that basketball involves a large number of movements in various directions (jumps, acceleration, deceleration, lay-ups, and cutting movements), which can impose on the athlete certain demands that can lead to injuries of the lower extremities. The Pedar system was used for extracting, from 10 plantar regions, two important parameters: peak pressure (PP) and pressure-time integral (PTI). The results showed that, when wearing softer shoes, the plantar loading was reduced in various regions of the foot for the basketball players during the four analyzed types of movement. Lower values of PP and PTI were found especially in the forefoot part, thus suggesting the use of a softer midsole in this region of the foot for reducing the high plantar loading experienced by the athletes. Further experiments are still necessary in order to evaluate if this solution may have any negative influence on the performance of the basketball players [94].

Another study [95] evaluated the extent to which foot loading is affected by the court surface (wood or asphalt) on which basketball is played. For this purpose, three tasks related to basketball were performed, both on wood and asphalt courts, by the study participants: layup, jump shot, and maximal effort sprint. During the movements, in-shoe plantar loading was recorded by using the Pedar-X system, at 100 Hz. Furthermore, from eight regions of the foot (hallux, lesser toes, medial forefoot, central forefoot, lateral forefoot, medial arch, lateral arch, heel), the peak force, normalized to body weight, was extracted. The results showed that the medial forefoot and the toes are most exposed to higher impact forces when basketball is played on wooden courts, as compared to asphalt ones. These findings oppose the common belief that force attenuation is easier to obtain on wooden courts, which are more compliant than asphalt courts.

Injuries related to plantar stress in male basketball players were also researched in another study, in terms of the plantar load variations during three different basketball cutting tasks (sideward, 45° and 90° cutting) that require a maximum effort. Cutting movements typically imply a sudden body deceleration, followed by an acceleration in a new direction. The Pedar-X system, placed between the shoe insole and the plantar surface of the foot, was used to record the distribution of the insole plantar load during these tasks, at a sampling frequency of 100 Hz. The insole was divided into nine regions, each with a corresponding foot mask: medial and lateral heel, medial and lateral midfoot, medial, central and lateral forefoot, great toe and lesser toes. Data analysis using the insole plantar pressure system involved the extraction of the maximum force (MaF) and the peak pressures (PP), both at each foot mask and at the entire foot. Differences were found between the plantar loads during the three cutting tasks, posing injury risks for the lower extremities of basketball players. Thus, the highest values for MaF and PP were found at the heel during the 90° cutting, whereas the highest MaF at the total foot was recorded during the sideward cutting. These results might be used for developing new regeneration techniques for injuries in basketball players [96].

Another study [97], involving basketball players, where the Pedar-X system was used, aimed to determine the characteristics of foot loading and the number of trials needed for obtaining stable plantar pressure measurements, both during lay-up (the most frequently method used by basketball players for scoring points). This number of trials was unspecified by earlier studies or was arbitrarily averaged to three or five. Pedar-X was employed to record, at a sampling frequency of 100 Hz, three plantar loading variables: PP (peak pressure), PF (peak force), and PTI (pressure-time integral). These variables were extracted both from the entire foot and from each of the eight regions in which the foot was divided: hallux, lesser toes, medial forefoot, central forefoot, lateral forefoot, medial arch, lateral arch, and heel. The results showed increased plantar loadings during the takeoff steps at the heel and upon landing at both the heel and forefoot regions. The minimum recommended number of trials in this study was eight. This number was found experimentally to be the minimum one in order to provide stable measurements of loading during lay-up, across all regions of the foot.

Another sport where the Pedar-X system was used is hockey. The biomechanics of ice hockey skating were analyzed in a study by an on-ice measurement approach. Although skating is an essential movement in ice hockey, its biomechanics are less studied, because of the difficulty of collecting on-ice data. Because of the lack of anterior studies evaluating the biomechanics of on-ice skating technique, by means of recording a combination of kinetic, kinematic, muscle activity, and insole pressure data, the new study proposed an approach able to collect these data in a comprehensive manner. Thus, in order to address this gap, for the proposed approach, its reliability was tested, and afterwards, it was implemented for researching both the forward skating technique (by means of the differences between two phases of forward skating, namely acceleration and steady-state) and the technique differences between various skill levels. The participants, both high and low caliber hockey players, wore a 3D accelerometer on the right skate, for stride detection, where the second stride was defined as acceleration (ACC) and the sixth stride as steady-state (SS). The Pedar-X system (Novel, Minnesota, USA), with the instrumented insoles with 99 sensors placed bilaterally into each skate, was used for recording the distribution of the plantar pressure inside the skate boot, at a sampling frequency of 90 Hz. The system also included EMG electrodes and electro-goniometers, which, together with the accelerometer, were directly connected to a data acquisition unit. The implementation was demonstrated to be successful and, moreover, identified the acceleration phase of forward skating as the most important factor in separating the performance of high and low caliber skaters [98].

Badminton is another sport that can benefit from the Pedar-X system measurement facilities. Thus, plantar load characteristics can be determined during the lunges performed by badminton players. Because long-distance powerful lunges are often performed by badminton players, a study [99] aimed to compare for the first time the plantar loads involved in one-step maximum forward lunges, in three directions, namely left-forward, right-forward, and front-forward lunges, each time wearing a different brand of shoe. The purpose was to prevent injuries in badminton players. The Pedar-X system was used for measuring insole plantar pressure in each case, namely three parameters: PP (peak pressure), MaF (maximum force), and CA (contact area). The foot was divided into nine regions: medial and lateral heel, medial and lateral midfoot, medial, central and lateral forefoot, great toe and lesser toes. The three parameters mentioned above were measured both for the entire foot and for each foot region. Data were collected only from the right foot (the landing one), at a sample frequency of 200 Hz. The study found higher plantar loads at the great toe region for the right-forward and left-forward lunges, as compared to the front-forward one, assessing thus the possibility of injuries in badminton players.

Rope skipping was also researched in terms of plantar forces’ evaluation with the Pedar system. Thus, a research group investigated the effects that athletic footwear may have on plantar force and ground reaction force during rope skipping. The participants skipped rope both with the one leg condition (OC) and two leg condition (TC). The Pedar-X system was used for collecting the values of plantar forces. The foot was divided into eight regions: rearfoot; lateral and medial midfoot; lateral, medial and middle forefoot; hallux; lateral toes. The measured parameters were PP (peak pressure), MaF (maximum force), FTI (force-time integral), PTI (pressure-time integral), and MPP (mean plantar pressure). The force platform AMTI was also employed, for measuring, on each skip, the vertical ground reaction forces (VGRF). The results showed higher values of all analyzed parameters for OC with both shoes employed in the study (a running shoe and a jumping shoe) as compared to TC. Similar values between the two shoes were obtained for the force distribution patterns. Plantar forces during the landing phase of rope skipping were found to occur mainly at hallux, lateral toes, and metatarsal head [100].

Soccer players might also benefit from the Pedar-X system. In a study [101], the maximum plantar force (Fmax) and contact time (CT) of the injured and uninjured limbs, both for those with less and respectively more than nine months of recovery after surgery (arbitrarily classified into these two groups), evaluated during moderate to fast speed running, were compared in soccer players who underwent ACLR (anterior cruciate ligament reconstruction) when passing the criteria that enabled them to return to sport (RTS). Both groups of soccer players after ACLR and healthy subjects ran at three different velocities on a treadmill. Pedar-X was used for measuring the plantar loading data, at a sampling frequency of 100 Hz, from both feet. Fmax was examined for the entire foot and was normalized to the body weight of each participant, for comparison purposes. CT was also evaluated for the entire foot, in absolute values. Although ACLR athletes with less than nine months after surgery met the RTS functional criteria, a relatively large asymmetry was observed in the vertical ground reaction force (vGRF) when running. This asymmetry was not observed, however, in the CT. When speed increases, the asymmetry also slightly increases, the reverse being true for athletes with more than nine months post-surgery and for the healthy controls.

### 6.3. Other Pedar Applications

Different running patterns were analyzed in various research efforts by using the Pedar-X system. One such study compared the plantar load characteristics for runners with two different strike patterns: RFS (rearfoot strike), the most common pattern, found in 75% of long-distance runners, and NRFS (non-RFS), including midfoot strike and forefoot strike runners (23% and 2%, respectively). NRFS running patterns are generally more beneficial in preventing running-related injuries (RRI), their incidence being 52.4% in RFS runners and only 22.8% in NRFS ones. The study aimed to find whether the running speed affects the plantar load. Plantar loads, when the right foot touched the ground, were measured, at a sampling frequency of 100 Hz, using the Pedar-X insole system, linked to the Pedar-X box, which was fixed to the participants’ waist. For the entire foot and also for nine regions, selected according to previous studies, five loading parameters were determined: PP (peak pressure), PTI (pressure-time integral), CA (contact area), MaF (maximum force), and FTI (force-time integral), the last two being normalized to the body weight of the participants. The study concluded that runners tend to adjust their CA according to the running speed, for the forefoot and midfoot regions. High impact forces are experienced by RFS runners on the midfoot and heel and by NRFS runners in the first metatarsal region [102].

Another study analyzing different running patterns evaluated a method of reducing the plantar pressure by increasing the preferred step rate during running, which is a strategy commonly used in managing RRI. In the study, the participants ran on a treadmill at five step rates (preferred, +5%, +10%, −5%, and −10%). The Pedar-X system was used in each case for collecting the plantar pressure data, at a sampling frequency of 100 Hz. The results showed that an increase of 10% of the preferred rate step during running reduces the plantar pressure at the midfoot and rearfoot, with no effect at the forefoot; a decrease of 10% increases the plantar pressure at the rearfoot, also with no effect at the forefoot; whereas the 5% increases and decreases have no effect on plantar pressures in any foot region. In conclusion, the authors recommended using a 10% increase of the step rate with beneficial effects on reducing the load on midfoot and rearfoot [103].

Different walking patterns were also analyzed using the Pedar-X system. A power spectral density (PSD) model was recently developed, based on experiments and normalized to walking frequency and order of harmonics, for the walking load of pedestrians. The proposal was made in the context of the effect that the movement of pedestrians may have when they walk on slender structures, like long-span floors or footbridges, causing the apparition of intense vibrations. The experiment was performed on a rigid floor. The Pedar-X system was used for measuring the plantar pressure data and estimating the vertical force and center of pressure for each foot of fifty-six test subjects. A force plate was also employed, and the Pedar system was calibrated by means of a comparison between the measurements from both the force plate and Pedar. The PDS model was successfully developed using a larger database than in previous models [104].

The estimation of the CoP parameter, both for walking and running, was the goal of other studies. Thus, the gait parameter is important because it is the point where the resultant force acts. In [68], this parameter was estimated from a limited number of pressure sensors, using the Pedar-X system. The authors of the study proposed to observe if the accuracy of CoP estimation is lowered, in walking and running conditions, when a smaller number of sensors is used. Thus, CoP was first calculated with all the 99 sensors of the Pedar-X system, at a sampling frequency of 100 Hz, in order to obtain the golden standard. Afterwards, eight different layouts were selected, as in Figure 6, containing odd numbers of sensors, from three to 17, chosen from among the 99 Pedar-X sensors. CoP was calculated for each case and compared to the gold standard.

One result of the study was that the root mean-squared errors (RMSE), calculated for evaluating the differences between the gold standard and each of the eight layouts, increased with the decrease of the number of sensors used, more significantly during running, thus limiting the accuracy in estimating CoP. These results might be useful when designing a low-cost insole system with a reduced number of sensors.

Another research work concerning CoP evaluation with Pedar-X consisted of using this insole system for determining CoP trajectory during gait, more precisely during its stance phase, in healthy adult persons [105]. The Pedar-X system was used for measuring the CoP coordinates on both transverse (X) and vertical (Y) axes. The authors chose to divide the stance phase into four sections, in which the position, velocity, and acceleration of CoP were determined. CoP was found to present a medial shift on the X axis and a forward shift on the Y axis. The study concluded that these results can be used as a standard norm when evaluating CoP trajectory in healthy adults. The Pedar system was also used in a study that evaluated the effects of medial wedge insoles when walking on surfaces with different geometries and hardness [106]. Plantar pressure and rearfoot movement were investigated in order to understand the interaction between the insoles and the surfaces. For this purpose, twenty-eight subjects with normal feet were asked to walk, with and without insoles, on a flat surface and upstairs, on all combinations of hard and soft surfaces. To evaluate if there is any effect of a 10° medial wedge insole on a hard or soft walking insole, a sandwich of two full-length medial wedge insoles was used together with a neutral running shoe. The Pedar-X in-shoe system was used to measure the vertical plantar pressure. An electrical goniometer, mounted on the heel, was used for measuring the rearfoot movement of the left foot. The synchronization between Pedar-X and the goniometer was done using the sync device from the Pedar-X recorder software. Around 2800 steps were loaded into the database. The maximum force, contact time, and force-time integral were calculated. For each step in the database, the following parameters were determined: take-off (TO), velocity (Vel), maximum angle (Max), minimum angle (Min), touch-down (TD), and static measurement. The results showed that the variation of the surface hardness did not induce significant changes on the medial wedge insoles, thus attesting to their effectiveness.

The placement of sensors differs in the various research attempts. Therefore, a study aimed at determining the optimal placement of the Pedar-X system sensors when walking [107]. For this purpose, healthy subjects, both male and female, wore experimental shoes and socks that included, for each foot, the 99 pressure sensors of the Pedar-X system, and afterwards, they walked on a treadmill, at a 3 km/h constant speed, for approximately one minute. The sampling frequency was chosen as 50 Hz. Pearson correlation analysis was used for determining the correlation coefficients between the sensors’ positions and the subjects. The results showed that the highest values of the correlation coefficient were found at the following four regions, ordered from higher to lower values: heel, metatarsals, toe and finger line, and barefoot outline regions, respectively. Such an analysis might be extended from simple walking to running, jumping, walking up and down stairs, etc.

Another recent application of the Pedar system was for improving the precision of measurements of body weight (BW). The purpose of the study was to evaluate the Pedar insole for BW measurements, both under static and dynamic conditions, and to validate the measured dynamic forces without needing additional costly and complicated devices and setups. In the experiments, a participant, wearing Pedar-X, was asked to stand on a Kistler force plate (in order to measure his BW) and then to perform six motion activities, in the following order: walking at a slow, medium, and fast speed; running at a medium and fast speed; and ending with self-induced limping (used as a comparison to the other gait patterns). The dynamic and static BW values obtained from the Pedar system were calculated and compared to the value obtained from the Kistler force plate. By applying a baseline correction, a decrease of the error of 4% (from 6% to 2%) was obtained for the Pedar system accuracy, thus demonstrating a method for assessing the accuracy of the devices that measure plantar pressure, without the need for other equipment, and also for improving the efficiency of these devices [63].

An interesting study used the Pedar system for assessing the effect of heel heights on the postural control of women, when they stand on a dynamic support surface exposed to sinusoidal oscillations [108]. The results of a 2003 survey by the American Podiatric Medical Association [108,109] showed that between 37% and 69% of women choose to wear high heels in their daily life, with a height of the heels that generally exceeds 5 cm, forcing the foot to remain in the position pf plantar flexion, thus disturbing the normal position and function of the ankle joint and increasing the probability of injuries.

In [108], a study including college women who had previously worn high heels for at least two years was presented. The Pedar system (with 99 sensors and approximately 2.6 mm thick insoles) was used for measuring the distribution of the plantar pressure (by evaluating the pressure-time integral, the contact area, and the coordinates and trajectory of the CoP). The subjects were asked to remain standing in the center of a movable platform, with feet apart at shoulder-width, and to try to maintain their equilibrium without stepping. The variables included were two independent ones: the heel heights (0.8, 4.2, and 6.6 cm) and the movement directions of the platform (three different ones). The plantar surface was divided into seven regions: big toe (BT), other toes (OT), medial forefoot (MFF), center forefoot (CFF), lateral forefoot (LFF), middle foot (MF), and heel (HEEL). The results of the study showed that, for healthy young women, shoes with high heels alter the mean position of the COP toward the medial forefoot (MFF), mainly because of morphological changes (the arch rising that makes the foot shorten immediately when placed in a high-heeled shoe). This result was verified by the variations of the other two parameters measured in different regions with Pedar (contact area and pressure-time integral). A lower tolerance to sudden perturbations was also observed when wearing high-heeled shoes [108].

Another study [110] assessed the effect of high-heeled shoes (HHS) on the stability of the feet of the wearers during quiet standing. Four heights of the shoes were chosen: 1, 5, 8, and 10 cm, respectively. The effects were evaluated on three parameters: mean CoP, variability (standard deviations) of mean CoP, and CoP velocities, respectively, both in the ML (medial-lateral) and AP (antero-posterior) directions. Measurements of the plantar pressures were done by using the Pedar-X system, at a sampling frequency of 50 Hz. It was found that the postural stability is affected in these conditions, decreasing significantly in the ML direction (and slightly decreasing in the AP direction) with the increase of the heel height. The adverse effects were found to be increased for 8 cm HHS and the worst for the 10 cm heeled shoes.

## 7. Description of Proposed Methods Using Other In-Shoe Systems

Besides the above-mentioned applications of Pedar and other in-shoe commercial systems, some studies proposed other such systems, developed by the researchers themselves, for several applications. We shall analyze some relevant examples of such in this section.

BioFoot^®^ is an in-shoe system developed in 2001 by Biomechanics Institute of Valencia, Spain. The system is equipped with 64 piezoelectric sensors and can acquire samples with a sampling rate ranging from 50 Hz to 250 Hz [111]. It is provided with a wireless connection (11 Mbs), and data can be sent to and logged by a computer [112]. Recently, it was used in a study to determine the correlation between the plantar pressure variation, the shoe wear, and the comfort of runners with the same type of shoes [111].

Another in-shoe system for the measurement and analysis of plantar pressure was proposed, based on a textile fabric pressure sensor array [6]. The sensors are connected with the help of conductive yarns to a soft polymeric board and integrated into an insole, which is interfaced with a stable system for data acquisition, which transmits these data by means of Bluetooth, to the remote receiver. In order to select the sensor position, the sole of the foot was divided into 15 regions (as in Figure 7): heel (Areas 1–3), midfoot (Areas 4–5), metatarsal (Areas 6–10), and toe (Areas 11–15).

For reducing the complexity of the system, six positions were selected for the first prototype shoe (at the heel and metatarsal areas, because these areas have higher pressure both for children and adults during normal activities), as shown in Figure 8. The following parameters were determined by using a real-time analysis software: CoP, peak pressure, mean pressure, and shift speed of CoP. The advantages of the system are that it is soft and light, having a high-pressure sensitivity and a long service life, and possessing stable performance in both static and dynamic measurements [6].

Another research group developed an in-shoe device, sensitive to pressure, for monitoring the distribution of plantar pressure, with the purpose of gait analysis. The device consisted of a matrix with 64 sensitive elements (based on an optoelectronic transduction principle), on-board electronics, and a battery. Each sensitive element consists of a light emitter (LED) and a light receiver (photodiode). It offers a 100 Hz data acquisition frequency and wireless data transmission, and the autonomy in continuous functioning mode is, on average, 7 h. The device was experimentally characterized and validated preliminary on a healthy subject [77].

An in-shoe system for monitoring plantar pressure was also designed for assessing the exercise stress in daily activities [113]. The proposed system (shown in Figure 9) is composed of four parts: an insole equipped with force sensors (sensorized insole), a microcontroller (for signal acquisition and transmission), a radio interface based on Bluetooth communication technology, and a power unit.

The insole contains five force sensors, which are able to estimate with a reasonable accuracy the foot load. The system is capable of performing, in real time, parameter measurement, transmission, and data analysis and storage. It has the advantages of being portable, user-friendly, low power, and low cost.

Another research group [114] designed and evaluated a customizable array of pressure sensors, able to be trimmed into various sizes. The array consists of flexible circuits and piezoresistive, uniformly-distributed pressure sensors made of textile material. The flexible circuits include wires that ensure that, after the array is trimmed, one can still use the remaining full or partial sensors. For the evaluation part of the study, gait parameters were analyzed for the designed insole shape arrays. The results showed that, irrespective of trimming or lack thereof, the spatial and temporal patterns of pressure distribution were acquired successfully for the proposed customizable array of pressure sensors.

## 8. Challenges and Open Issues

There are several challenges faced by both smart sock systems and in-shoe systems such as Pedar-X. Only a few of them are shared by a majority of the proposed designs and architectures. Generally, each of them has its own difficulties and perspectives for enhancing its performance.

However, there are a few recurrent issues. In order to be able to be used on a larger scale, their price is still too high. Their efficiency could be increased, together with their comfort for the user. A more robust system, capable of measuring larger intervals of the parameters, is also important. For a more detailed and precise measurement, a higher number of sensors is necessary. For in-shoe systems, the number of sensors should not be less than 15, with maximum dimensions of 5 mm × 5 mm [13], for enabling an accurate measurement without the need to increase the energy consumption and the computational cost. Furthermore, the sampling rate should be higher (for Pedar-X, it is limited to a maximum of 100 Hz). The weight of the system is required to be no more than 300 g [13], but future improvements might also increase the weight supported by such systems. Last, but not least, the number of participants in most of the studies was limited, and especially in medical applications, many of the proposed devices were not tested on real patients. Thus, larger samples of the specific population at which the devices are targeted should be taken into consideration in future studies, in order to achieve a validation of their performance in the respective applications.

In the following, we shall analyze the main challenges and open issues faced by some of the representative systems reviewed in this paper.

SmartSox [50], the device based on optical fibers proposed for managing risk factors involved in diabetic foot amputation, has three main limitations: insufficient spatial resolution, lack of precision in measuring skin temperature, and too few characteristics determined (as future work, four types of determinations were proposed: diabetic foot ulceration prediction, classification of foot type, identification of the foot at risk, and footwear effect assessment).

The smart sock system proposed for monitoring skin temperature in [10] still has to determine the number of steps of the participants; the effect of room temperature needs to be assessed in future studies, as well as the effect of BMI (body mass index) on skin temperature.

Six challenges were found in the analyzed study concerning the proCover system, proposed for prosthetic limbs’ sensory augmentation by means of covers made from smart textile materials [47]. Thus, there is a need for better connectors (for reducing the necessary time for putting on the smart socks); the durability of the textile material must be verified; long-term effects should also be considered; components should meet three criteria: smaller, wireless, and able to be used with an external power supply.

Algorithm enhancement is one of the open issues in [30], together with the necessity to also calculate other posturographic parameters using Sensoria socks, the reproducibility verification, the extension of the studies in different cases with closed eyes, and finally, the use of various smart sock devices to discriminate between normal and pathological results.

One of the studies that admitted the need of more participants is [39], as well as a higher number of analyzed activities in order to be able to perform a classification. This research concerning step counting with smart socks also envisioned improvements both at the hardware level (smaller board; less power consumption; using Bluetooth Low Energy for data transmission) and the software level (other classification algorithms, faster than SVM, used in the study; other classifiers, such as decision trees or artificial neural networks).

Furthermore, for step counting applications with smart socks, reference [31] suggested the necessity of comparing the results with those obtained with other pedometers; a better recharging method; and the inclusion in the tests of not only healthy persons, but also ill and elderly ones.

For measuring the abnormal relative plantar pressure with a battery-free smart sock [35], the authors proposed two future additions: a smartphone application, to accompany the smart sock and to both enable a graphic visualization of the foot pressure and to evaluate the health problem; and a solution for minimizing the recharging time of the system, by a better harvesting of energy, together with storage designs that permit higher rates of monitoring and a wider application domain.

The smart socks proposed in [9], for a continuous monitoring of the temperature, in diabetes patients, might benefit from three directions of new research: including a higher number of persons, for the study to be statistically significant; developing an optimum reporting manner (concerning four aspects: content of the report; frequency of notifications; preventing the development of foot ulcers; reducing the amputation cases); and most importantly, a built-in tracking activity, involving the use of data from the sock for monitoring the activity of the patients.

The P-SOCK smart wearable oximeter developed by [58] seeks the improvement of the device so that it can be used both in home and NICU environments; attaching a device for oxygen therapy; the display of the parameters on the screen of the device; creating an application on the mobile phone for real-time monitoring of the parameters.

The gait segmentation method of [22] might be strengthened by using a higher sample of participants and by evaluating not only the contribution of the plantar pressure on gait, but also of the one of the upper body.

In their application of gait partitioning with smart socks, the work in [26] proposed four future directions: higher sample of participants, better electronics, improved sampling rate, and the comparison of the obtained results with a reference.

The same research group [25] suggested, for improving their method of detecting excessive pronation and supination with smart socks, a larger sample of participants, including not only healthy persons, but also patients, and finally, in order to improve the quality of the results, an increase of the sampling rate, by means of improved hardware solutions.

Smart socks based on EMG [46] might benefit from other classification algorithms that prove to be more efficient and by increasing three factors: hardware performance; battery lifetime, and the impermeability level of the system.

The position of sensors on the insole in a Pedar-X study [107] that was analyzed only for walking may also be tested in various other situations, such as running, jumping, walking up and down stairs, etc.

For determining the center of pressure, such as in [68], different positions and dimensions of the sensors should also be taken into account for a better accuracy of the results.

The application of Pedar-X for basketball players [95] might evolve mainly three new directions: obtaining also kinematic data, such as sprint speed or jump height; evaluation of the stiffness for various court surfaces (besides wood and asphalt, used in the study); taking into account not only the plantar forces, but also the joint kinetics.

For badminton applications, the work in [99] proposed extending the sample of participants to include not only right-handed males, but also left-handed ones and women; taking into consideration the court surfaces and the designs of the shoes; evaluating the biomechanics of the inferior limbs also in other sports.

For applications of Pedar-X system in evaluating the changes of plantar pressure in HRS shoes, the authors of a recent study [91] proposed two upgrades: the inclusion of patients in the tests, along with the healthy participants; the estimation of kinematic data concerning the knee, ankle, and hip, in order to be able to evaluate the gait stability and comfort of the users.

For patients with low back pain, the work in [89] proposed taking into account, as future directions, other speeds of walking, selected by the participants themselves; evaluation of the differences in results between males and females; analysis of the possible influence of the thickness of the adipose tissue on the outcome. For patients with KOA (knee osteoarthritis), the work in [86] suggested a reduction of the weight of the system, accompanied by tests performed on real KOA patients, in order to determine the efficiency of the device.

For patients with KOA as well, the work in [87] proposed five improvements: a higher sample of participants; data taken from many walking trials, not only one; analysis of other characteristics as well, such as knee alignment or KL-grade (Kellgren–Lawrence classification); evaluation of the longer term effect on the plantar forces; and the combining of the plantar pressure measurements with the gait analysis.

Other systems, such as the Footscan platform [80], may also benefit from a higher number of participants in the study; from tests performed not only on healthy volunteers, but also on ill ones; an analysis could be performed additionally in order to see if the obtained results may also be applied to other systems.

Furthermore, the Zebris system [62] studies should also include a larger sample of participants, including—as is the case in the analyzed comparison with Pedar—not only healthy males, but also the elderly, children, and females, at various speeds of walking and running; a higher sampling rate; a more frequent calibration of the Pedar-X system, on a daily basis (not only at the beginning of the study).

## 9. Conclusions

In this paper, we presented the state-of-the-art for smart socks and in-shoe systems, with a particular focus on the Pedar-X system. We reviewed a large range of applications for both technologies, in gait monitoring, and in applications in domains such as medicine, sports, and various other specific areas.

Firstly, we presented the general aspects regarding gait monitoring systems and the motivation behind them. In addition, we presented the concept of plantar pressure and the applications in which this parameter is mainly used. After presenting the mandatory features of such systems and devices, we classified them into four main categories (platform systems, in-shoe systems, smart wireless insoles, and smart socks).

Secondly, we emphasized medical applications as Parkinson’s disease, diabetes, and various disorders where smart socks and insole systems represent a solution to the problem of detecting in time the abnormal health conditions. Moreover, we considered the field of sensory augmentation of prosthetics, which comprises useful applications dedicated to patients missing their limbs. Solutions in this area can increase the quality of life of amputees, through haptic feedback.

Thirdly, the Pedar system was described and compared to other commercial solutions, at it is a benchmark in this area. Often, given the same reason, the proposed systems in the literature are compared to Pedar.

Throughout the paper, we reviewed both commercial products and research-driven systems. Moreover, we highlighted the main strengths of the proposed devices and also their limitations and future improvement directions.

The vast literature review in this domain was covered for the first time, to the best of our knowledge, in a coherent and comprehensive manner. Researchers in this field are, thus, given a useful tool for selecting relevant information and developing new designs and systems, also extending the research towards other smart textiles and in-shoe system applications using sensors.

## Figures and Tables

**Figure 1 sensors-20-04316-f001:**
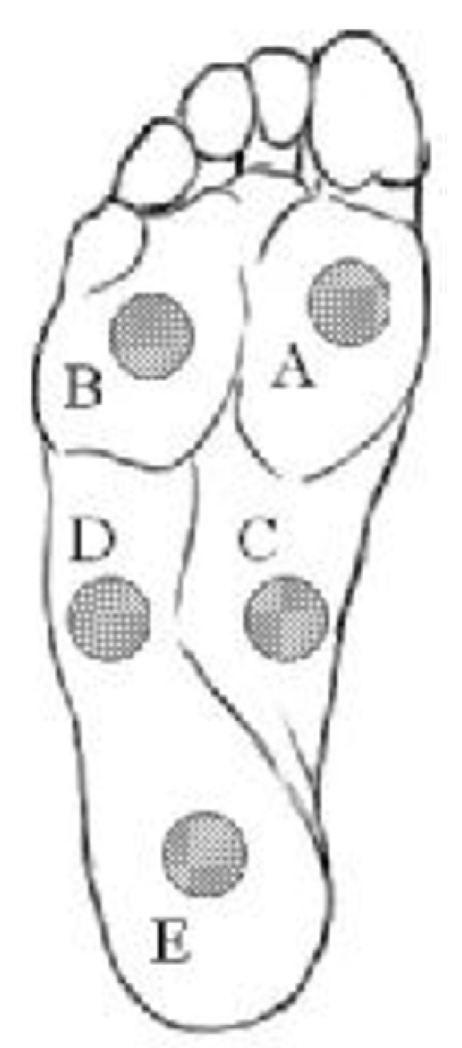
Placement of the smart sock sensors [17].

**Figure 2 sensors-20-04316-f002:**
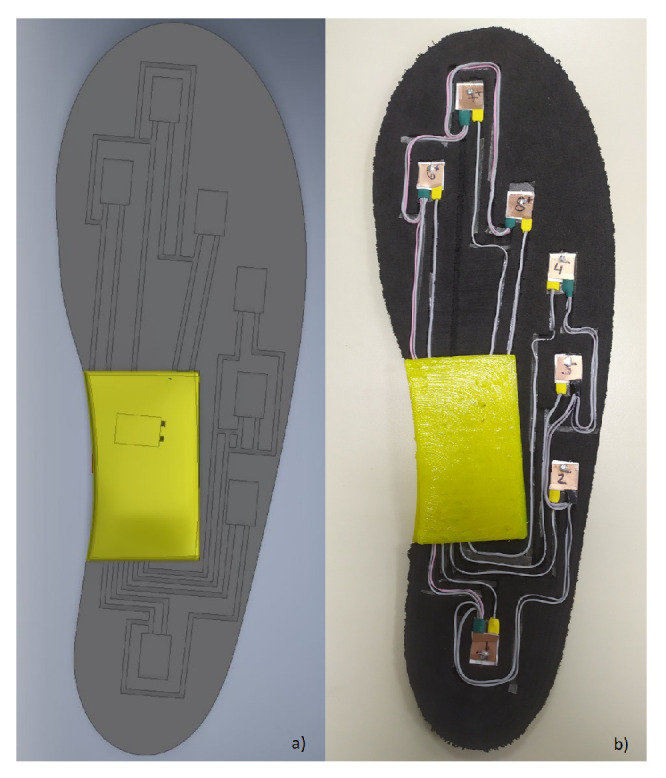
Arrangement of the sensors in the smart sock proposed by [22]: (**a**) 3D design; (**b**) inside view of the insole.

**Figure 3 sensors-20-04316-f003:**
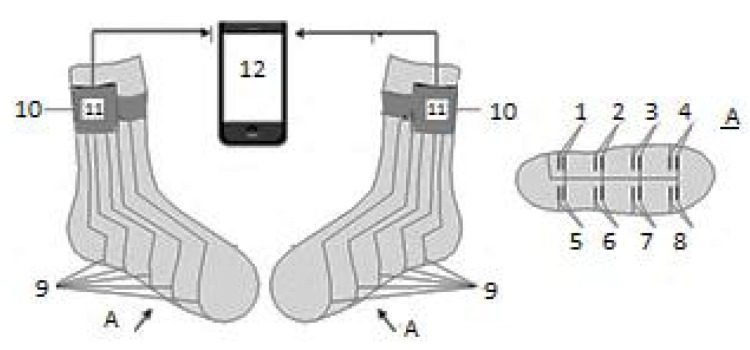
DAid^®^ Pressure Sock System (DPSS) [2]. 1–8: pressure sensors; 9: conductive lines; 10: connectors; 11: data gathering and forwarding component; 12: data processing component.

**Figure 4 sensors-20-04316-f004:**
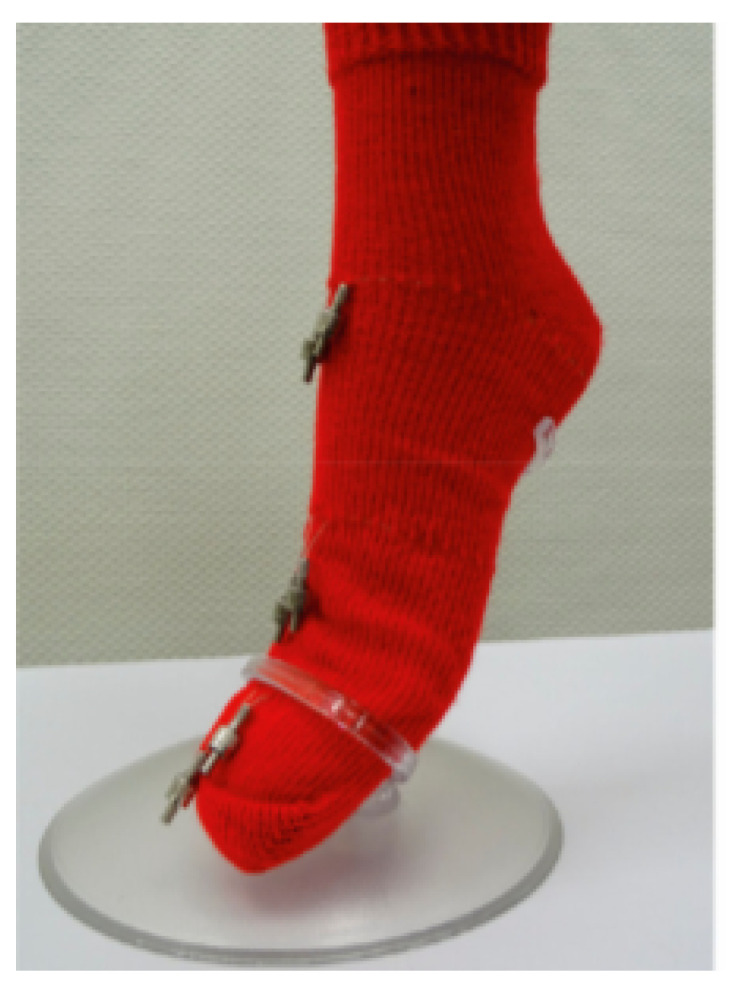
POF-based knitted sock [38].

**Figure 5 sensors-20-04316-f005:**
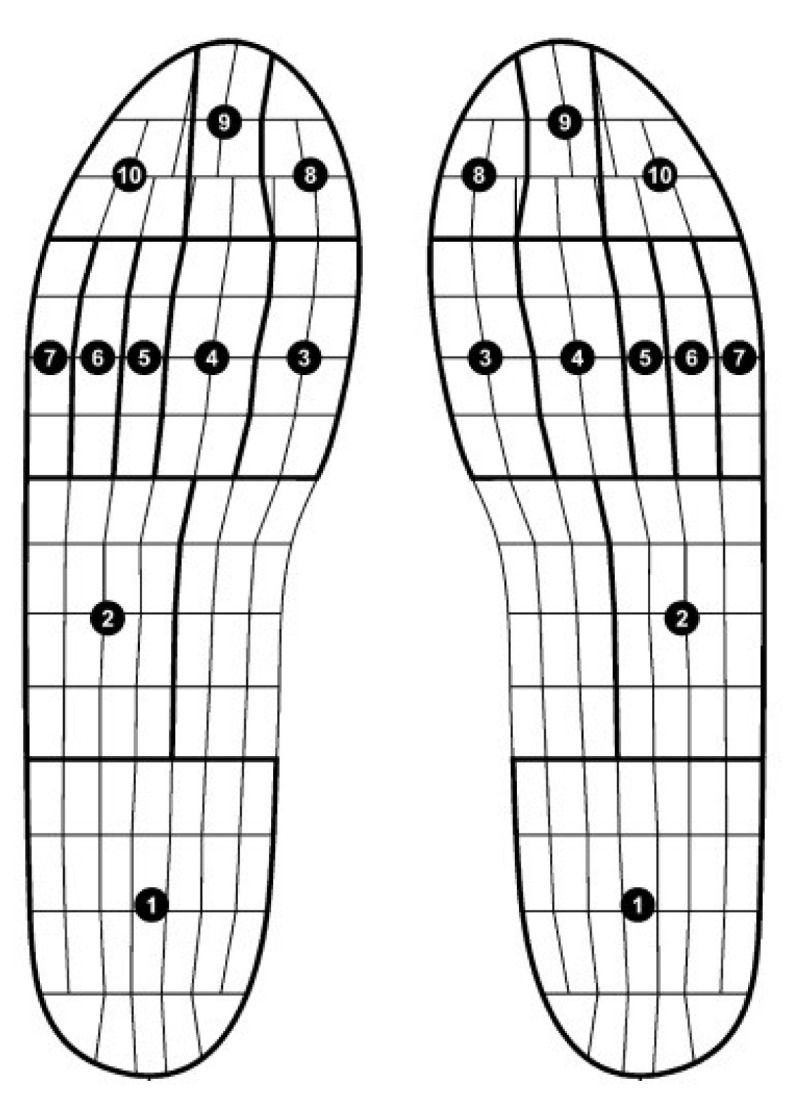
The left and right insoles used for Pedar, divided into 10 regions [67].

**Figure 6 sensors-20-04316-f006:**
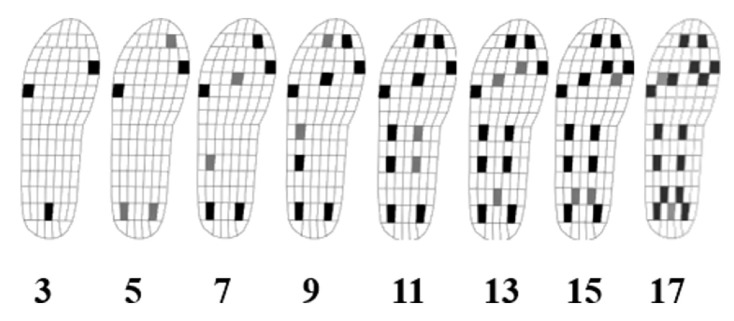
The eight different layouts, with 3, 5, 7, 9, 11, 13, 15 and 17 sensors, respectively proposed in [68] for measuring CoP with reduced number of sensors.

**Figure 7 sensors-20-04316-f007:**
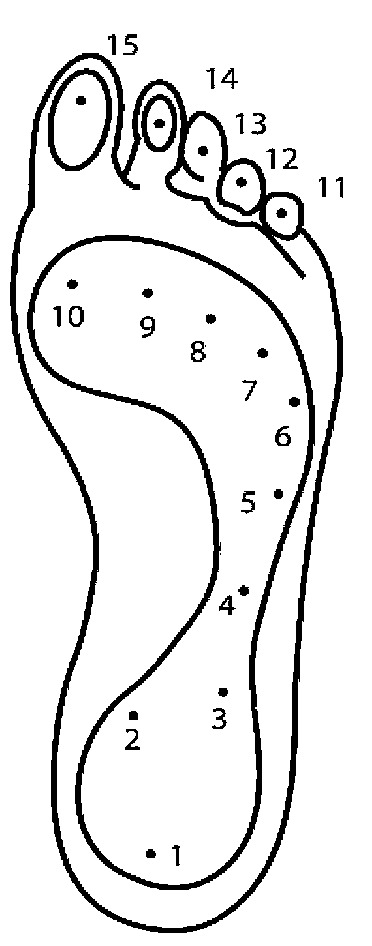
Division of the foot sole into 15 regions [6].

**Figure 8 sensors-20-04316-f008:**
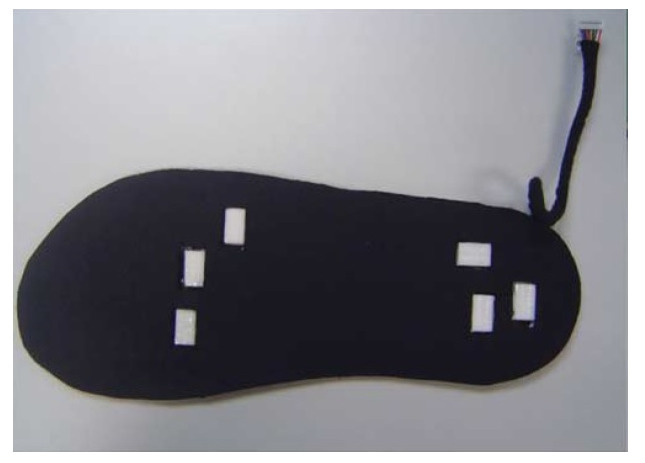
Insole with six textile sensors [6].

**Figure 9 sensors-20-04316-f009:**
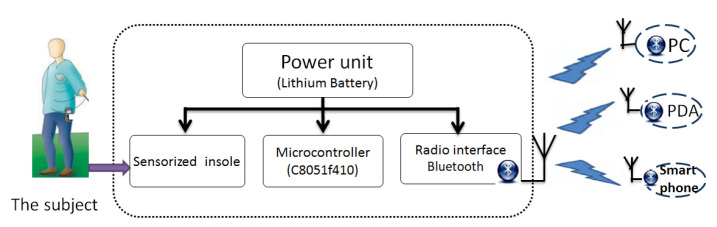
Structure of the system proposed in [113].

**Table 1 sensors-20-04316-t001:** Features of gait monitoring systems.

Feature	Description	Motivation
Wearable	Implies low-weight devices,	Performing measurements in different environments
	wireless technologies	and conditions, not only in the laboratory
Accurate	Uses accurate and reliable devices	Reliable data and new measurements in the same
		scenarios and conditions with the same output
Comfortable	Implies imperceptible casing;	Avoiding disturbing the user and performing
	it is secured against accidental detachment	erroneous experiments
Safe	Implies appropriate isolation against	Avoid injures and fatalities
	electrical discharges and ground loops	

**Table 2 sensors-20-04316-t002:** Smart sock research initiatives for gait and foot pressure analysis.

System	Application	Method	Sensing (Type/No.)	Communication Technology	Reference
Smart sock	People suffering from gait disorders	Conductive thread, placed between a neoprene and a conductive fabric	Resistive textile pressure sensors (polyester-BASF resistant carbon fibers)/6	Wired (serial data logging)	[23]
Smart sock	Gait analysis	Comparison of conductive textiles in terms of their sensing ability	Multiple piezo-resistive sensor patches	WiFi	[24]
Smart sock	Analysis of gait parameters	Algorithm for distinguishing heel strike and non-heel strike walking and running modes	Resistive sensors knitted in the sock/5	Bluetooth	[17]
Smart sock	Detection of excessive pronation and supination of the foot; gait cycle partitioning	Values given by the sensors are converted into a pressure vector	Piezoresistive sensors/5	Bluetooth	[25]
Smart sock	Gait cycle partitioning	Algorithm with six gait phases	Piezoresistive sensors/5	Bluetooth	[26]
Smart sock	Gait cycle partitioning and gait parameters’ determination	Algorithm for segmentation of the gait cycle and for gait parameters determination	Capacitive pressure sensors/8	Bluetooth	[22]
Sensoria smart sock and smart shirt	Differentiation between normal and abnormal gait	SVM, ANN, LDA, and kNN	Pressure sensors/3 and accelerometer/1	wireless (no CoT mentioned)	[27]
Sensoria smart sock and smart undershirt	Discrimination between three different postures (lying down, sitting, and standing) and various walking and running activities, with different speeds	ANN, LDA, and kNN	Pressure sensors/3 and accelerometer/1	wireless (no CoT mentioned)	[28]
Sensoria smart sock	Gait monitoring	Measurement of step count, velocity, and cadence	Pressure sensors/3 and accelerometer/1	wireless (no CoT mentioned)	[29]
Sensoria smart sock	Posturographic assessment	Variations of CoP parameter evaluation	Pressure sensors/3 and accelerometer/1	wireless (no CoT mentioned)	[30]
Sensoria smart sock	Counting steps in slow walking	Three different methods, using: (1) a smart sock worn on the left foot; (2) a pedometer; (3) a pedometer included as an application in a smartphone	Pressure sensors/3 and accelerometer/1	Bluetooth	[31]
Algorithm to implement in Sensoria smart socks	Finding frailty phenotypes	Algorithm using an artificial neural network	Gyroscope/1	Bluetooth	[31]
DAid^®^ Pressure Sock System (DPSS)	Gait analysis for normal and flat foot	Plantar pressure measurement	Piezoresistive pressure sensors/8	Bluetooth	[1,2,32]
DPSS	Testing of shoe cushioning properties	Plantar pressure measurement	Piezoresistive pressure sensors/8	Bluetooth	[33]
Version of DPSS	Gait parameters measurement	Processing, analysis, and representation of gait parameters during outdoor walking and running; foot loading during gait is compared to the propagation of a shock or seismic wave	Piezoresistive pressure sensors/6	Bluetooth	[14,34]
Battery-free smart sock	Detection of abnormal changes of relative plantar pressure values	Measurement of relative plantar pressure	Piezoresistive pressure sensors/4	RFID reader unit, two antennas oriented orthogonally	[35]
SWEET-Sock	Postural and gait analysis	Measurement of parameters for postural and gait analysis	Piezo-resistive textile sensors/3 and accelerometer/1	Simblee BLE (Bluetooth Low Energy)	[36]
GRPS (ground reaction pressure sock)	Determination of the ground reaction pressures	Sensors are placed on top of a BodiTrak vector plate, positioned in turn on a Kistler force plate	Compressible soft robotic sensors (C-SRS)/10	BLE	[37]
E-knitted POF-based sock	Measurement of friction during walking	Irradiance loss evaluation	Empa Geniomer^®^ POF/3	N/A	[38]
Smart sock	Counting steps	The smart socks gather information concerning motion and the degrees of ankle bending; three algorithms are used: for classification, step counting, and interaction with the user	Accelerometer/1, magnetometer/1, gyroscope/1, bending sensors/4.	wireless (no CoT mentioned; Bluetooth mentioned as future research)	[39]

**Table 3 sensors-20-04316-t003:** Comparison between the DAid^®^ and OptoJump systems.

Locomotion Type	Walking	Race Walking	Running
**Absolute mean difference (s)**	0.0027	0.0024	0.0013

**Table 4 sensors-20-04316-t004:** DAid^®^ validation with respect to the BTS system.

System	DAid^®^	BTS
**Mean ground contact time (s)**	0.281	0.298

**Table 5 sensors-20-04316-t005:** Comparison between Zebris and Sensoria systems.

System	Zebris	Sensoria
**Sway Path (mean ± SD) (mm)**	868 ± 81	884 ± 71
**Mean Sway Velocity (mean ± SD) (mm/s)**	14 ± 2	9 ± 1

**Table 6 sensors-20-04316-t006:** Smart sock research initiatives.

System	Application	Method	Sensing (Type/No.)	Communication Technology	Reference
Temperature sock	Temperature foot monitoring in diabetes	Measuring foot temperature, alerting	IC-based and NTC temperature devices	wireless (no CoT mentioned)	[7]
Texisense smart sock	Plantar ulcer prevention	Tissue overpressure notification	Texisense pressure sensing fabric	Bluetooth	[41,42,43]
Temperature sensing socks	Smart textiles, diabetes ulcerations	Temperature monitoring and decision-making	Temperature sensing yarn	wireless (no CoT mentioned)	[8]
Smart sock wireless device	Foot temperature monitoring (diabetes and neuropathy)	Detection of abnormal increase of temperature based on measurements performed every 10 s	Neurofabric™textiles based on temperature microsensors/6	Bluetooth	[9]
Smart sock	Foot ulceration prediction	Study correlation between increased skin temperature and plantar pressure overload	Thermal sensors (NTC thermistors)/7	N/A	[10]
Distal EMG sock	Body control, fall detection	Distal EMG signal feature estimation	EMG sensors/5 conductive electrode pairs, 6 wet electrodes pairs	N/A	[18]
Smart wearable sock	PLMD detection	Monitoring the activity of PLM related muscles	sEMG system with Nishijin electrodes/2	N/A	[44]
Wellness assessment sock	Wellness statistics	Points-based score using sensors data	HR/HRV, FSR, temperature, GSR, SpO${}_{2}$ sensors, accelerometer	WiFi	[19]
Instrumented Sock	Drop foot, gait events’ identification	Kinematic signals derivation based on video camera	resistive strain sensors	Wired	[45]
MONARCA	Bipolar disorder signs’ recognition	Physical and social activities and behavior recognition based on sensors and smartphone data	Smartphone sensors (GPS, accelerometer), wrist-worn sensor (accelerometer, gyroscope), smart sock (GSR, pulse sensor)	Bluetooth	[20]
Smart EMG-based socks	Age-related gait changes, fall risk and postural anomalies’ detection, sarcopenia	Linear discriminant analysis	Myoware muscle sensor/2	Bluetooth	[46]
proCover	Sensory augmentation for prosthetic	Sensing and haptic feedback	EeonTex LG-SLPA fabric	N/A	[47]
Self-functional sock	Energy harvesting-based wearables, sports, healthcare	Single electrode mode gait analysis, walking pattern detection, and motion tracking	Hybrid mechanism for sensing devices: piezoelectric and triboelectric	N/A	[48]
MagicSox	Drop foot detection	Classification normal foot/drop foot based on support vector machine and multiplication of backward differences	FlexiForce A201 (Tekscan) piezoresistive pressure sensor/1, flex sensors/2, gyroscope/1, accelerometer/1	Bluetooth	[49]

**Table 7 sensors-20-04316-t007:** Smart sock commercial developments.

System	Application	Method	Sensing (Type/No.)	Communication Technology	Reference
SmartSox	Foot ulcer parameters’ assessment	Sensors data processing to extract joint angles, temperature, and pressure variation	Optical fiber sensors/5	N/A (LabVIEW interface only)	[50]
Owlet Smart Sock	Baby monitoring	Pulse and oxygen levels monitoring	Pulse oximeter	WiFi (base station use is possible), Bluetooth	[51,52,53,54]
Baby Vida	Baby monitoring	Pulse and oxygen levels monitoring	Pulse oximeter	WiFi (no base station use is possible), Bluetooth	[52,53]

**Table 8 sensors-20-04316-t008:** Components of smart socks according to [42].

Component	Role
Sock (textile)	Foot external pressures sensing and acquisition
Central unit	Gathering data and forwarding to external device
External device	Data processing and information extraction for estimating foot ulcer risks in the patient

**Table 9 sensors-20-04316-t009:** Pedar parameters used to evaluate repeatability in [67].

Parameter	Acronym	Measure Unit	Parameter	Acronym	Measure Unit
Peak Pressure	PP	kPa	Pressure-Time Integral	PTI	kPa·s
Contact Area	CA	cm${}^{2}$	Force-Time integral	FTI	N·s
Contact Time	CT	ms	Instant of Peak Pressure	IPP	ms

**Table 10 sensors-20-04316-t010:** Validation results of proposed smart insoles with respect to the Pedar system and force plate (FP) in terms of CoP accuracy [61].

Participants	CoP	*p*-Value	SI vs. FP		Pedar vs. FP	
			$R^{2}$	k	$R^{2}$	k
1	CoPx	0.0989	0.7046	0.6655	0.6825	0.7458
	CoPy	0	0.9077	0.8455	0.9401	1.08
2	CoPx	0	0.7837	0.8867	0.8409	1.0492
	Copy	0.0001	0.9368	0.8538	0.9244	0.9053

**Table 11 sensors-20-04316-t011:** Comparison between the most important features of the Medilogic, OpenGo, Tekscan, and Pedar in-shoe systems. Developed from [69,70] and the manufacturers’ specifications [71,72,73].

Feature	Medilogic	OpenGo/Insole3	Tekscan	Pedar
Pressure sensor model	SohleFlex Sport	Moticon proprietary	FScan 3000E Sport	Pedar-X
System cost (current quote)	11,600 €	1795 €/7500 €	15,500 €	14,000 €
Price including	insoles	insoles/insoles+software	insoles	-
Pressure sensor technology	Resistive	Capacitive	Resistive	Capacitive
Number of pressure sensors/insole	Variable based on insole size (up to 240)	13/16	Variable based on insole size (up to 960)	99
Pressure sensor density	0.79 per cm${}^{2}$	0.1 per cm${}^{2}$	3.9 per cm${}^{2}$	0.57–0.78 per cm${}^{2}$
Other sensors	-	3D accelerometer/3D accelerometer+3D gyroscope	-	-
Communication technology	WiFi	2.4 GHz ANT/BLE5.0	wired, wireless	Bluetooth, fiber optic/TTL
Analysis Software	medilogic	Beaker/Moticon Science	F-Scan	Pedar
Insole thickness (at sensor region)	1.6 mm	2–3 mm	0.2 mm	2.2 mm
Maximum sampling rate	300 Hz	50 Hz/100 Hz	169 Hz	100 Hz
Measurement range	6–640 kPa	0–400 kPa/0–500 kPa	345–862 kPa	20–600 kPa
Calibration method	By manufacturer (polybaric characteristics)	No calibration needed	Device: factory insole: human standing or calibration device	Insole: Tru-Blu (pneumatic calibration)
Recommended time between calibrations	1 year or 5000 steps	-	Disposable insoles, calibrate at each use	Variable

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
