# Peer review of "Smart Socks and In-Shoe Systems: State-of-the-Art for Two Popular Technologies for Foot Motion Analysis, Sports, and Medical Applications"

_sensors, 2020, doi:10.3390/s20154316_

Round 1

Reviewer 1 Report

This manuscript entitled “Smart socks and in-shoe systems: State-of-the-Art for Two Popular Technologies for Foot Motion Analysis, Sport and Medical Applications” aimed to review the most recent advances in research concerning two popular devices used for foot motion analysis and health monitoring: smart socks and in-shoe systems. But there are several questions should be addressed before this manuscript can be accepted for publication, which lists below.

Specific comments

  1. Line 9-13, “From the extensive literature research … using sensors in smart textiles and in-shoe systems.”, This is not a good sentence construction and it is unclear. Please modify this sentence.
  2. Line 14, Avoid using “smart socks”, “in-shoe systems” and “medical applications” in the keywords, since it appears in the title.
  3. Line 18, “…and to train the foot to detect sudden falls…”, This sentence is not clear. How do you train your feet to detect falls? The authors should consider revising this sentence to avoid confusion for the readers.
  4. Line 20&23, Please remove the bolding for “foot pressure” and “temperature”.
  5. Line 21-22, “Most studies …. There are also a few other studies …”, Please add references.
  6. Line 50, “a maximum number of five step pairs”, The maximum number of steps may depend on the type and number of Platform Systems. The “five step pairs” may not be rigorous enough.
  7. Line 107-119, These two paragraphs seem not relevant to this part (General considerations concerning gait monitoring).
  8. Line 123-124, “…that they are able to monitor the contact between ground and the feet soles.”, monitor the contact between the foot sole and the shoe or between the ground and the foot sole?
  9. Line 138, “Thus”, please revise.
  10. Line 141-142, “… more powerful processing must be employed”, What is “more powerful processing”?
  11. Line 325-326, “This is an advantage … that were uncomfortable and complicated.”, This sentence needs to improve.
  12. Line 401-403, “thus assessing the very good temporal characteristics of the DPSS smart sock.”, What dose “temporal characteristics” mean? The accuracy of the temporal parameter? “good” is not suitable to be used because it is subjective.
  13. Line 463, “A small group of normal subjects participated in another smart sock study…”, Authors should consider using specific number instead of “A small group of”.
  14. Line 489, “CoP” rather than “centre of pressure (CoP)”.
  15. Line 560, based on Figure 5, the readers may mistake 2 for the lateral midfoot.
  16. Line 568-569, Authors should consider adding comparative validation results.
  17. Line 587&592, Please keep the abbreviations consistent. COP or CoP.
  18. Line 612, “The results were promising, for applying the system to facilitate gait retraining.”, This sentence needs to be improved.
  19. Line 719-722, These sentences are not relevant to this paper. The author should consider simplifying them.
  20. Line 734-735, Once again, the authors provide full name and its abbreviation (center of pressure (CoP)). The authors should recheck the manuscript to avoid such problems.
  21. Line 751-753, Please add a ref.
  22. Line 825-826, “The minimum recommended number of trials in this study was eight”, How do they got this minimum number of trials?
  23. Line 947-948, “The collected data … stored on an SD card.”, This sentence is not necessary.
  24. Line 981, “The results of a 2003 survey by the American Podiatric Medical Association showed that…”, Please add the corresponding reference.
  25. Please check the text in Figure 9 (red wavy line).
  26. Line 1184, “…that we consider useful”, Please consider revising.

Author Response

Dear Reviewer,

Thank you!

Reviewer 2 Report

The authors present a considerable amount of works here. This must be mentioned here. However, the authors drown under the information they would like to share, at the expense of overall quality of the manuscript. First, the chosen form does not help the content, and it is too often looking like a catalog of information, sometimes not enough supported by reference or detailed results. There is also a lack of critical work by the authors, what is highly requested in such a state of the art.

  1. General considerations concerning gait monitoring

The authors must consider rewriting all this section. The content is fine, but the ideas are chained unrelated and makes reading uncomfortable. Authors jumps from few bullet points about the possible interest for plantar pressure to consideration about what should be the characteristics for a wearable sensor, then list various system to capture pressure, comeback to technical specification, and end with consideration about running/walking without special link with the previous paragraph.

Also, an introduction about the purpose of the article is required somewhere within the first section.

Page 1, Lines 17-19: If the message can be understood, this introductive sentence is not written in an optimal manner. Decompose at least in two parts, with sports, then medical application.

Page 1, Lines 23-24: “This parameter is closely linked with pressure, but it is sometimes considered more efficient, since it is able to be more accurately measured than pressure.” Add a reference for this statement.

Page 2, Line 45: “generally embedded in a walkway” should be in the floor or a treadmill.

Page 2, Line 49: “and only for measurements on the bare foot, not also on the shod foot.” This statement is wrong, it is possible also to measure the pressure under the shoes. What is not possible is to get information about pressure at the interface foot / shoe.

Page 2, Line 50: “they are able to measure only a small number of steps (a maximum number of five step pairs).” Again, incorrect statement. Most systems installed in clinical settings are platforms embedded in treadmill. The limitation for step count is not valid.

Page 2, Line 53: “More flexible, with a higher efficiency and mobility”. The efficiency indicates the way the inputs are used by the system. For which purpose in-shoe systems would be more efficient than platforms?

Page 3, Line 70: “For platform systems, the most used commercial product is EMED®-SF platform (Novel Inc., USA)”. From which statistics or reference? In which countries? I could cite a dozen different systems without searching and, for example in my country, Novel EMED is not widely used.

Page 4, Line 107: Again, a jump from technology to running specificities, then walking, for which explanation are at least imprecise shortcuts or limited point of view, if not incorrect.

  1. Description of proposed methods using smart socks in special applications

Add an introduction at the start of this section for the reader to understand what you will investigate here. I was surprised then to read section 3 was about gait analysis while several points in section 2 involved gait analysis (for example from lines 252). Make a major distinction between both sections.

There is a lot of interesting information here, but again the lack of organization and common thread makes the whole text difficult to follow. It seems here the reader must read one section of your paper and reorder him/herself the content to get the essence of your work. That’s normally the job of the authors…

Page 4, Line 125: “a number of five sensors was proposed, that can be placed in different parts of the sock and are able to determine motion’s spatio-temporal parameters”. I am not certain that standards or recommendations should be based on Proceedings from a non-specialized conference.

Overall, the section provides interesting information, but without clear line in the ideas. Wanting to cover all application of the smart socks, the authors often just oversee and create a list of unrelated information. For example, try to group the applications which are most closed each other. 

  1. Description of proposed methods using smart socks for gait and foot pressure analysis

Page 11, Line 343: “running (with a velocity of 6 km/h)”. Transition speed from walking to running typically occurs around 2.0 m/s (7.2 km/h), 6 km/h is too slow to be running.

Figure 2 and “gait partitioning” (Page 11, Line 362): Is there no limitation or validation flaws to discuss? I cannot imagine how to analyze locomotion with such a location of backward and forward sensors. E is too central in the heel and there no forward sensors.

Page 11, Line 368: “The recently proposed partitioning involved six gait phases”. From the onset of 2000 and even before, it is recent compared to Antiquity…

Page 12, Line 382-388: “All three devices showed step times with practically equal values, thus validating the proposed smart sock system.” Purpose about concurrent validity like what is described here should be documented with better details and values. What is the level of adequacy of “practically equal”?

Page 12, Lines 390-395: What sort of validation the authors are referring to? Again, reference is from a conference. We all know that this does not warrant a good review of the quality of the presented work. Always prefer finding reviewed articles.

Page 13, Line 442: “One such device was targeted towards the detection of abnormal relative plantar pressure changes.” and all the paragraph. Is this whole paragraph related to the reference [53] written line 462? If yes, it must be stipulated at the beginning of the paragraph or it is hard to state if about the level of this information.

Page 14, Line 463-468: Same comments than previously. It is of primary importance for the reader to know what has been tested and detailed results, not only “The authors concluded that Sensoria smart socks are a reliable alternative to Zebris and, moreover, benefits of a lower cost.” Also, this leads me to wonder if the authors of the current paper did look at any conflict of interest related to the considered product in all “validation” papers they listed in this manuscript?

Page 15: Lines 498-502: Activities recognition should be better in section 2.

  1. Description of Pedar system, together with validation and repeatability tests for Pedar and other in-shoe and platform systems

Page 16, Lines 548: “Pedar® mobile system is currently considered the gold standard among in-shoe systems and is one of the most used and well-established sensor insole pressure measurement systems [16]”. This could be true but cannot be based on one reference. If it is so well-established, no difficulty to find a lot of valuable references.

  1. Description of proposed methods using in-shoe Pedar system

Overall, this section is better written than the previous sections, even if there is always a trend to jump between topics through the section.

  1. Description of proposed methods using other in-shoe systems

That is ok for this section, but the content is more focused and limited (in a good meaning).

  1. Challenges and Open Issues

Finally, we got the brainstorming section! Unfortunately, it reflects the main structure of the manuscript.

Page 28, Line 1055: “Thus, better algorithms are required, for analysis and classification, also a comparison between them and their performance could be performed.” This is too less: please summarize from your work what are the flaws of current algorithms. One sentence on this, then consideration about the “price is still too high” … it is so frustrating.

Page 28, Line 1072-1076: What is the interest of developing a lot of metrics if these metrics are based on imprecise or limited measurement. Would be better to propose few high quality and validation parameters than 100 invalid outcomes.

Author Response

Dear Reviewer,

Thank you!

Round 2

Reviewer 1 Report

The authors have made a good revision, it is suitable to be accepted. 

Reviewer 2 Report

Authors made substantial improvement to the initial submission. It is clear that modifying the structure of the manuscrit had yet concurred to a better picture of the whole information and therefore help the reader to get the message. All validation details and introductive parts improve the scientific level and contribute to get an article form instead of a master thesis.

If in my point of view, some parts could always be improved and always consider that two separated papers would have been better; but it is also based on personal point of view and feelings. On the other hand, authors proposed a detailed and complete content, try to follow advises and answer comments, and this article should provide new information or at least unique article summarizing these information.

For these reasons, I will anyway consider this could be accepted for publication in Sensors.